# PRISM: Demystifying Retention and Interaction in Mid-Training

**Bharat Runwal** [1 2]   **Ashish Agrawal** [1]   **Anurag Roy** [1]   **Rameswar Panda** [1 2]

## Abstract

Mid-training is increasingly used to improve the reasoning capabilities of large language models (LLMs), yet its design choices and interaction with evaluation and reinforcement learning (RL) remain poorly understood. Prior work often focuses on narrow domain gains, overlooking retention of general abilities, long-context performance, and RL compatibility. We present **PRISM** (Demystifying Retention and Interaction in Mid-Training), a holistic empirical study that analyzes mid-training design choices, what to evaluate, and how domain mixtures and training stages interact across model families. Experiments on Granite-3.3 8B, Granite-4 family (Micro and Tiny variants with both dense and hybrid architectures), LLaMA-3.1 8B, and Mistral-7B/24B base models show that a relatively small, high-quality mid-training phase of $\sim$**27B** tokens acts as a critical stabilizing stage for reasoning. Across models, **PRISM** yields consistent gains of $\sim$**6–10** points on coding benchmarks and $\sim$**17–30** points on mathematical reasoning benchmarks while preserving general performance. RL applied on top of **PRISM**-mid-trained models produces stable, monotonic improvements, adding a further $\sim$**3–8** points across coding and math tasks such as LiveCodeBench, Codeforces, AIME, and MATH500, whereas RL applied directly to base models is substantially less effective. Our results demonstrate that retention-aware mid-training is a necessary intermediate step for reliable reasoning enhancement and RL scaling, and provide practical guidance for designing robust mid-training pipelines for modern LLMs. **Project page available at:** https://bharat-runwal.github.io/PRISM/

[1]IBM Research [2]MIT-IBM Watson AI Lab. Correspondence to: Bharat Runwal <bharatrunwal@gmail.com>.

*Proceedings of the $43^{rd}$ International Conference on Machine Learning*, Seoul, South Korea. PMLR 306, 2026. Copyright 2026 by the author(s).

## 1. Introduction

The training of Large Language Models (LLMs) has followed a now-familiar trajectory: large-scale pre-training on web-scale corpora followed by task-specific alignment phase of supervised fine-tuning (SFT) and RLHF. Recent large-scale models, introduced an additional intermediate stage of training *mid-training*, in which higher-quality or domain-focused mixtures are used to imbue high quality knowledge in pre-trained checkpoints before downstream alignment and RL (Team et al., 2025a; Olmo et al., 2025).

Despite its growing adoption, *mid-training* remains loosely defined in the literature. Some works treat mid-training as a long-context extension phase that focuses on architectural/context capacities (Abdin et al., 2024), others use it as a higher-quality, annealing stage to add capabilities and domain knowledge (improving on math-specific benchmarks like GSM8k) (OLMo et al., 2025), and a recent line of work investigates mid-training choices that explicitly prepare models for subsequent post-training stages by incorporating instruction-following data and chain-of-thought traces (Thinking data) (Wang et al., 2025). These different usages have converged in practice, but the field lacks a holistic study that systematically quantifies the trade-offs induced by mid-training design choices (data mixtures, evaluation strategies, and downstream RL).

In this work, we present **PRISM** (Demystifying Retention and Interaction in Mid-Training) Fig. 1, a broad empirical study designed to analyze the structure of mid-training decisions and their effects across model families. Our experiments span multiple open-source model families (Granite, LLaMA, and Mistral) and a comprehensive evaluation suite covering general LLM performance, long-context, coding, mathematical reasoning, and science benchmarks, including LB-V1/V2, Ruler, LiveCodeBench, Codeforces, AIME, MATH500, and GPQA-Diamond.

Using targeted mid-training mixtures of only $\sim$**27B high-quality tokens**, we study (i) how mid-training affects retention of general abilities and long-context behavior, (ii) how domain mixtures interact (Math+Code+Science synergy), (iii) how to combine mid-training with RL, and (iv) how mid-training effects scale across model sizes.

Our key findings are:

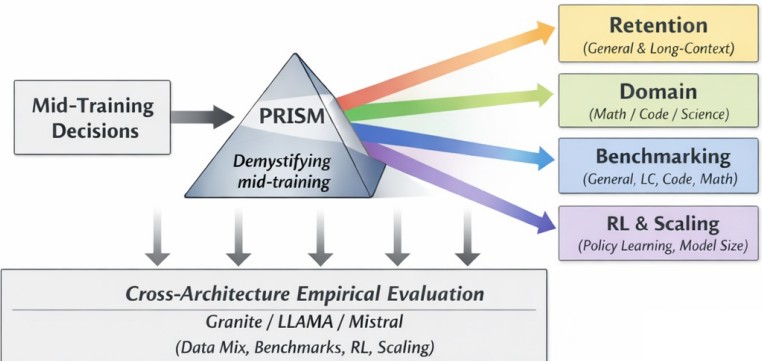

*Figure 1.* **PRISM overview.** Mid-training decisions are decomposed into their principal design axes, including retention of general and long-context abilities, domain interaction (math, code, science), benchmark selection, reinforcement learning compatibility, and scaling behavior. PRISM enables holistic evaluation of mid-training choices across model families at scale.

- A relatively small, high-quality mid-training phase (∼**27B** tokens) produces large, robust gains in reasoning: across tested models we observe roughly ∼**6–10 point** gains on coding benchmarks and ∼**17–30 points** on math benchmarks while preserving general performance.

- Applying RL on top of PRISM-mid-trained models yields stable, monotonic improvements, an additional ∼**3–8 points** on coding/math and substantial gains on science (GPQA-Diamond), whereas RL applied directly to base models is often less effective and less stable.

- Domain mixtures interact non-trivially: Math+Code mixtures produce synergistic improvements for both code and math tasks, while other domains (like science) can have neutral or task-specific effects; careful composition is necessary for balanced mid-training.

- The beneficial patterns generalize across model families (**Granite, LLaMA, Mistral**) and persist across reasonable scaling regimes.

## 2. Limitations of Prior Mid-Training Approaches

> **Takeaway:** Prior mid-training work often delivers domain-specific gains at the cost of generalization and holistic evaluation, and is rarely coupled with broad benchmark analysis or controlled studies of downstream RL behavior.

Recent mid-training strategies for large language models (LLMs) have demonstrated notable improvements in targeted capabilities such as coding and mathematical reasoning by introducing higher-quality or domain-focused data between pre-training and downstream fine-tuning or reinforcement learning (RL) stages (Olmo et al., 2025; Wang

et al., 2025). However, the term *mid-training* has been used inconsistently in the literature, referring to a diverse set of practices including long-context extension, data annealing, and domain-specific capability refinement, without a unified framework or standardized evaluation benchmarks.

A key limitation of existing approaches is their narrow evaluation focus. Many studies report gains on a limited set of domain-specific benchmarks (e.g., math or code) without assessing whether these improvements preserve general-purpose language modeling capabilities or interact with other reasoning dimensions (Wang et al., 2025). For instance, long-context extension work primarily evaluates context-window scaling and retrieval-style tasks, with limited analysis of its impact on general reasoning or downstream performance (Abdin et al., 2024). Similarly, domain-focused mid-training recipes often emphasize improvements on math or code benchmarks while omitting broad generalization and cross-domain robustness evaluations (OLMo et al., 2025; Wang et al., 2025).

Another shortcoming is the lack of controlled investigation into how mid-training interacts with downstream optimization, particularly reinforcement learning. While prior work suggests that certain mid-training strategies can facilitate RL by better aligning representations with downstream objectives, these claims are typically evaluated within narrow experimental settings and lack systematic comparison across model families, domains, and benchmark suites (Wang et al., 2025; Zhang et al., 2025).

Concurrent work has begun to address parts of these gaps. Recent studies show that mid-training can serve as a distributional bridge between pre-training and post-training, reducing distributional mismatch while preserving general capabilities (Liu et al., 2025). Other work develops controlled experimental frameworks that isolate the respective contributions of pre-training, mid-training, and reinforcement learning (RL) to reasoning generalization, highlight-

| Dataset | Type | Tokens (B) |
|---|---|---|
| DCLM-EDU | General web data | 111.46 |
| Open-R1 (MoT) | Math reasoning | 0.60 |
| Nemotron Post-Training v1 | Math | 35.93 |
| Megamath-Web-Pro | Math web | 14.73 |
| Open-R1 (MoT) | Code reasoning | 1.18 |
| OpenCodeReasoning-2 | Code reasoning | 1.12 |
| RefinCode | Code web | 186.44 |
| StarCoder2 | Code web | 432.73 |
| Open-R1 (MoT) | Science reasoning | 0.42 |
| OpenThoughts3 | Science reasoning | 0.73 |
| WildChat-1M | Chat | |
| Tulu-3 SFT Personas | Chat | 0.91 |
| UltraChat-200k | Chat | |

*Table 1.* Datasets used in mid-training mixtures. Token counts are reported in billions (Granite 3.3, 8B).

ing mid-training as a critical yet underexplored stage across model families (Zhang et al., 2025). However, these analyses are primarily conducted at small scale, for example on models such as Qwen2.5-100M, leaving open questions about whether the observed trends extend to larger scale.

Taken together, these limitations motivate PRISM: a retention-aware empirical framework that evaluates mid-training choices across multiple domains, benchmark axes, and downstream RL behavior across model families to uncover trade-offs overlooked by prior work.

## 3. Data mixtures and preprocessing for mid-training

> **Takeaway:** Mid-training performance is highly sensitive to data composition; carefully tuned mixtures that balance general web and instruction data with domain-specific reasoning sources yield robust retention and consistent gains, and we adopt these empirically validated splits across all experiments.

**Datasets:** For the Math and Code domains, we use two data types: general web documents and domain-specific reasoning datasets. General web data helps retain knowledge acquired during earlier phases of pretraining, while reasoning datasets imbue domain-specific problem solving ability. For the Science domain, we include only reasoning-focused datasets.

We apply lightweight, deterministic preprocessing to all datasets to ensure data quality and evaluation integrity, including filtering out low-quality web documents, selecting verified reasoning examples, and normalizing conversational formats. Dataset-specific filtering and formatting details are provided in Appendix Section A.

Table 1 summarizes the datasets used for mid-training. Prior work such as OctoThinker (Wang et al., 2025) shows that

incorporating a small amount of general instruction data can stabilize reinforcement learning; accordingly, we also include general instruction-following datasets in our mid-training mixtures. However, OctoThinker focuses primarily on math-domain in mid-training and downstream RL. In contrast, our goal is to support general reasoning across diverse domains while retaining knowledge acquired during earlier pretraining phases. To this end, we include both general web data (e.g., DCLM-EDU) to preserve broad capabilities and domain-specific web and reasoning datasets to induce targeted improvements.

Fig. 2 reports the final per-source sampling weights we use for mid-training. After experimenting with various weightings across domains, we found the configurations in the table to provide the best balance between retaining broad pretraining knowledge and inducing targeted domain improvements; consequently, we adopt these splits as the default sampling policy for all experiments reported in this paper.

## 4. Benchmark decisions

> **Takeaway:** Evaluate mid-training with a *balanced* suite that measures (i) general LLM ability, (ii) long-context behaviour, and (iii) domain-specific reasoning; otherwise, domain gains may mask regressions.

In PRISM we adopt a deliberately broad evaluation setup to surface both gains and regressions introduced by mid-training. Concretely, we combine general leaderboards (LB-V1 and LB-V2) with focused long-context, code, math, and science evaluations so that improvements in a single domain cannot hide capability loss elsewhere.

**Practical guidance when selecting benchmarks.** As summarized in Table 2, effective evaluation of mid-training decisions requires both breadth and depth across reasoning dimensions:

- **Mix breadth and depth:** combine general-purpose leaderboards (LB-V1 and LB-V2) with targeted domain benchmarks to expose global regressions while accurately measuring domain-specific gains.
- **Measure long-context retention explicitly:** evaluate long-context reasoning separately (e.g., RULER), as mid-training dominated by short-context data can degrade long-context capabilities, often necessitating an additional lightweight fine-tuning stage to recover performance (see Section 7.1).

Overall, a robust mid-training evaluation should jointly include (i) broad leaderboards to detect regressions in general capability, (ii) focused domain benchmarks to quantify gains, and (iii) explicit long-context and instruction-

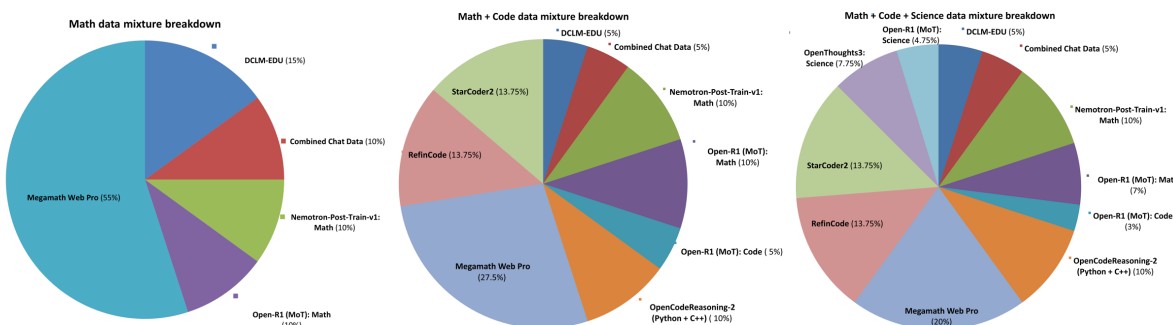

*Figure 2.* Mid-training data mixture configurations and per-source sampling percentages.

| Category | Benchmarks | What it measures | Why it matters |
|---|---|---|---|
| General ability | Leaderboard-V1 (LB-V1) (ARC, HellaSwag, MMLU, TruthfulQA, Winogrande, GSM8K), Leaderboard-V2 (LB-V2) (IFEval, BBH, MATH, GPQA, MUSR, MMLU-Pro) | Broad multitask knowledge and robustness | Detects generalization regressions hidden by domain-specific gains. |
| Long-context | RULER | Long-context retrieval | Ensures mid-training does not degrade long-context retrieval capabilities. |
| Code | LiveCodeBench, Codeforces | Executable program synthesis and reasoning | Captures real-world coding ability and avoids contamination. |
| Math | GSM8K, AIME, MATH500 | Multi-step mathematical reasoning | Highly sensitive to data quality and mid-training composition. |
| Science / QA | GPQA-Diamond | Expert-level scientific reasoning | Probes domain expertise and Google-proof reasoning. |

*Table 2.* Benchmark categories recommended for evaluating mid-training design choices.

following checks to ensure downstream alignment and reinforcement learning remain effective.

## 5. Domain-wise Effects of Mid-Training Data

> **Takeaway:** Mid-training performance is driven by data composition. Domain-specific data delivers large gains in its corresponding benchmarks, while balanced mixtures across math, code, and science achieve the best overall trade-off, improving domain reasoning while preserving general capabilities.

Having established the data sources and empirically validated mixture configurations used for mid-training in Section 3, we now examine how the inclusion of domain-specific data affects downstream performance across benchmarks. We use the Granite-3.3 (8B) base model and mid-train the same model using three progressively richer data mixtures: Math-only, Math+Code, and Math+Code+Science, following the configurations in Fig. 2. All mid-training experiments are conducted with a fixed budget of ∼**27B** tokens at a context length of 8192; additional training hyperparameters are provided in Appendix Section B. We evaluate each resulting model on both general-purpose leaderboards (LB-V1 and LB-V2) and domain-specific benchmarks for mathematics, coding, and scientific reasoning. This setup allows us to isolate the effect of intro-

ducing data from each domain and to analyze the resulting trade-offs between domain specialization and retention of general capabilities.

**Impact of math-focused mid-training.** Introducing math-specific data during mid-training leads to substantial improvements in mathematical reasoning. Compared to the baseline model, the Math-only mixture increases the Math average from 8.95 to **36.43**, a gain of **+27.48** points (Table 4). These gains demonstrate that high-quality math reasoning data is the primary driver of mathematical capability during mid-training.

**Effect of adding code data.** Adding code-specific data produces large improvements on programming benchmarks. While Math-only mid-training yields only marginal code gains over the baseline, increasing the Code average from 2.07 to 2.81 (**+0.74**), the Math+Code mixture raises the Code average to **10.71**, corresponding to a **+8.64** point improvement relative to the baseline (Table 4). Incorporating science data on top of code does not substantially alter code performance, with the Math+Code+Science mixture maintaining a similar Code average of **10.58**.

**Effect of adding science data.** Including science data during mid-training improves performance on GPQA-Diamond without deteriorating code or math performance. Compared to the Math+Code mixture, the Math+Code+Science mix-

| Mixture | Leaderboard V1 | | | | | | | Leaderboard V2 | | | | | | |
|---|---|---|---|---|---|---|---|---|---|---|---|---|---|---|
| | ARC | HellaSwag | MMLU | TruthfulQA | Winogrande | GSM8K | OpenLLM V1 Avg | IFEval | BBH | MATH | GPQA | MUSR | MMLU-Pro | OpenLLM V2 Avg |
| Base | 61.95 | **83.46** | 62.56 | **52.24** | **80.35** | 56.33 | 66.15 | **46.62** | 24.68 | 10.20 | 6.38 | 8.88 | 23.82 | 20.10 |
| Math only | **62.54** | 78.72 | **64.29** | 46.04 | 75.30 | 71.95 | 66.47 | 46.46 | 25.57 | 17.75 | 5.59 | 9.08 | **29.86** | 22.39 |
| Math + Code | 61.01 | 78.09 | 62.65 | 47.36 | 74.74 | 73.46 | 66.22 | 45.56 | **26.87** | 18.43 | 5.93 | **10.60** | 28.40 | 22.63 |
| Math + Code + Science | 61.69 | 78.12 | 62.98 | 46.96 | 74.90 | **74.22** | **66.48** | 46.44 | 26.32 | **20.02** | **7.27** | 8.60 | 29.55 | **23.03** |

*Table 3.* Leaderboard V1 and V2 results for Granite-3.3-8B mid-trained with the mixtures in Fig. 2

| Mixture | Code Avg | Math Avg | GPQA-D |
|---|---|---|---|
| Base | 2.07 | 8.95 | 22.56 |
| Math only | 2.81 | **36.43** | 17.34 |
| Math + Code | **10.71** | 44.33 | 19.02 |
| Math + Code + Science | 10.58 | **48.75** | **29.12** |

*Table 4.* Domain-specific evaluation results for Granite-3.3 (8B). Code Avg is the mean of LiveCodeBench and Codeforces; Math Avg is the mean of AIME 24, AIME 25, and MATH500. Full benchmark-level results are provided in Appendix Table 8.

ture increases GPQA-Diamond from 19.02 to **29.12** (**+10.10** points). At the same time, the Code average remains stable (10.71 to **10.58**), and the Math average further improves from 44.33 to **48.75** (Table 4). These results show that science-focused data can be added without sacrificing gains in other reasoning domains.

**Trade-offs on general benchmarks.** Mid-training introduces measurable trade-offs on general-purpose benchmarks. On Leaderboard V1, the Math-only mixture improves the overall average from 66.15 to **66.47** (**+0.32**), driven primarily by gains on GSM8K, while exhibiting regressions on benchmarks such as HellaSwag and TruthfulQA (Table 3). Leaderboard V2 averages increase monotonically with broader domain coverage, rising from 20.10 for the baseline to **22.39** for Math-only, **22.63** for Math+Code, and **23.03** for Math+Code+Science. Notably, overall general performance remains stable across mixtures, which we attribute in part to the consistent inclusion of general web data from DCLM-EDU in all mid-training configurations.

# 6. PRISM Effects Across Model Families

> **Takeaway:** Across model families, architectures, and scales, PRISM mid-training consistently improves reasoning performance. We observe gains of **+15 to +40 points** on math benchmarks, **+5 to +12 points** on coding benchmarks, and **+6 to +13 points** on science benchmarks, while maintaining competitive general-purpose performance.

In this section, we study the effects of PRISM mid-training across multiple model families and architectural choices. Our evaluation includes Granite-3.3 (8B), LLaMA-3.1 (8B), Mistral-7B and Mistral-Small-24B, and Granite-4 Micro (3B), all of which use dense Transformer architectures. We additionally consider hybrid architectures within the

Granite-4 family: Granite-4-H Micro (3B), which combines attention and Mamba layers, and Granite-4-H Tiny (7B), which further incorporates a mixture-of-experts (MoE) design. Additional architectural and training details are provided in Appendix Section B.

For most experiments, we perform PRISM mid-training at an 8k context length, which offers a favorable trade-off between computational cost and downstream performance. As shown in the ablation study (Section B.4), increasing context length beyond 8k yields moderate gains across benchmarks, motivating this choice.

Table 5 summarizes the impact of PRISM mid-training across this diverse set of model families and scales. Across all models, PRISM consistently improves mathematical, coding, and scientific reasoning performance, while changes to general-purpose leaderboards are smaller and more model dependent.

**Hybrid versus dense architectures.** Within the Granite-4 family, we observe different trends to PRISM mid-training across dense and hybrid variants. The dense Granite-4 Micro (3B) shows substantial gains, improving MATH 500 from 50.42 to **79.40** and LiveCodeBench from 0.24 to **10.87**. Hybrid models, including Granite-4-H Micro (3B) and Granite-4-H Tiny (7B), also exhibit large improvements. For example, Granite-4-H Tiny increases AIME 24 from 6.30 to **41.45**, AIME 25 from 5.20 to **31.09**, and MATH 500 from 35.87 to **87.02**. While these results suggest that hybrid architectures can effectively leverage mid-training signal, differences in pretraining data and model scale prevent a direct attribution of these gains to architecture alone.

**Consistency across model families.** PRISM mid-training yields strong improvements across all evaluated model families. Mistral-7B shows large gains on math and code benchmarks, with MATH 500 improving from 1.68 to **70.71** and Codeforces from 0.15 to **9.42**. Mistral-Small (24B) similarly improves MATH 500 from 26.92 to **80.80**. LLaMA-3.1 (8B) benefits as well, improving AIME 24 from 0.05 to **16.45** and LiveCodeBench from 0.00 to **6.09**. These trends demonstrate that PRISM is effective across distinct model families and training recipes.

**Effect of model scale.** Although larger models achieve higher absolute scores, PRISM delivers meaningful gains at all scales. Smaller models often exhibit larger relative

| Model | Variant | Leaderboards | | Code | | | Science | Math | | | |
|---|---|---|---|---|---|---|---|---|---|---|---|
| | | LB V1 | LB V2 | LCB | CF | Code Avg | GPQA-D | AIME24 | AIME25 | MATH500 | Math Avg |
| Granite-4 Micro (3B) | Base | 66.01 | 21.82 | 0.24 | 2.28 | 1.26 | 21.55 | 16.09 | 12.34 | 50.42 | 26.28 |
| | PRISM | 62.91 | 20.53 | 10.87 | 6.25 | 8.56 | 34.34 | 27.19 | 22.29 | 79.40 | 42.96 |
| Granite-4-H Micro (3B) | Base | 64.49 | 18.99 | 0.60 | 0.88 | 0.74 | 20.88 | 7.08 | 2.70 | 30.17 | 13.32 |
| | PRISM | 64.21 | 18.75 | 15.53 | 8.02 | 11.78 | 32.66 | 33.69 | 23.49 | 82.73 | 46.64 |
| Granite-4-H Tiny (7B) | Base | 67.41 | 20.89 | 1.67 | 3.46 | 2.56 | 21.21 | 6.30 | 5.20 | 35.87 | 15.79 |
| | PRISM | 65.25 | 21.33 | 14.58 | 9.27 | 11.93 | 33.16 | 41.45 | 31.09 | 87.02 | 53.19 |
| Mistral-7B | Base | 60.88 | 14.89 | 0.00 | 0.15 | 0.07 | 26.94 | 0.00 | 0.10 | 1.68 | 0.59 |
| | PRISM | 59.99 | 19.68 | 10.16 | 9.42 | 9.79 | 24.07 | 28.85 | 24.27 | 70.71 | 41.28 |
| Mistral-Small (24B) | Base | 74.98 | 27.29 | 0.00 | 0.29 | 0.15 | 22.55 | 0.78 | 0.73 | 26.92 | 9.48 |
| | PRISM | 69.52 | 27.42 | 10.03 | 10.08 | 10.06 | 22.05 | 32.91 | 27.34 | 80.80 | 47.02 |
| LLaMA-3.1 (8B) | Base | 62.76 | 14.09 | 0.00 | 0.07 | 0.04 | 20.20 | 0.05 | 0.15 | 6.51 | 2.24 |
| | PRISM | 65.21 | 21.46 | 6.09 | 5.45 | 5.77 | 21.04 | 16.45 | 19.32 | 73.47 | 36.41 |

*Table 5.* Base versus PRISM (Math+Code+Science) mid-training results across model families. Code Avg is the mean of LiveCodeBench (LCB) and Codeforces (CF). Math Avg is the mean of AIME 24, AIME 25, and MATH 500. All values are reported to two decimal places.

improvements, while larger models realize strong absolute gains without severe degradation on leaderboards. For instance, Mistral-Small (24B) improves MATH 500 by more than **+50** points while maintaining Leaderboard V2 performance, whereas LLaMA-3.1 (8B) improves Leaderboard V2 from 14.09 to **21.46**. Overall, these results suggest that retention-aware, multi-domain mid-training provides consistent benefits across parameter scales.

## 7. Ablation Studies

### 7.1. Restoring Long-Context Ability After Mid-Training

Mid-training is performed at an 8k context length, which naturally degrades long-context capabilities inherited from pretraining. In this section, we study practical strategies to restore long-context performance after mid-training using Granite-3.3 (8B). We evaluate two approaches: (i) directly performing a short long-context extension phase on the mid-trained checkpoint, and (ii) linearly *merging* the mid-trained model with the base model prior to long-context extension. For both approaches, we further compare training all parameters versus training only attention modules during the long-context phase.

Details of the data construction and preprocessing used for long-context restoration are provided in Appendix Section B.3. In particular, we augment the training data with code examples containing longer chains of thought, apply filtering to remove short-context samples, and use best-fit packing to efficiently construct long-context training sequences.

**Effect of mid-training on long-context performance.** Mid-training at 8k context length significantly degrades long-context capability. While the Granite-3.3 (8B) base model achieves a RULER score of 59.09 at 128k context, the Math+Code mid-trained model drops sharply to **6.46**, de-

spite strong performance at short context lengths (89.02 at 8k). This confirms that mid-training with short-context data alone disrupts long-context behaviors learned during pre-training, motivating the need for explicit restoration strategies.

**Restoring long-context via long-context extension.** Performing a short long-context extension phase substantially recovers long-context performance. Applying 1k steps of long-context training directly on the mid-trained model raises RULER at 128k from 6.46 to **36.32** when training attention modules only, and to **38.41** when training all parameters. These improvements are consistent across intermediate context lengths, with RULER at 64k improving from 11.71 to over 54.63. At the same time, downstream reasoning performance is preserved or improved: Code Avg increases from 10.71 to **19.65** (attention-only) and **25.52** (full), while Math Avg remains above 42 across both variants (Table 6).

**Benefits of merging with the base model.** Merging the mid-trained model with the base model prior to long-context extension yields the strongest recovery at long context lengths. With a 15% base and 85% mid-trained linear merge followed by long-context training, RULER at 128k improves further to **42.16**, narrowing much of the gap to the base model. Importantly, this approach maintains strong downstream reasoning performance, achieving a Code Avg of **25.54** and a Math Avg of **44.48**. Across strategies, full-parameter long-context training yields the strongest recovery, while attention-only training still provides meaningful RULER improvements with competitive downstream performance, offering a practical efficiency–performance trade-off.

### 7.2. Effect of Mid-Training Token Budget

We next study the effect of increasing the mid-training token budget while keeping the context length fixed at 8k and using a Math+Code data mixture (Table 7). All experiments

| Model Variant | RULER | | | | | Code / Science | | | | Math | | | |
| --- | --- | --- | --- | --- | --- | --- | --- | --- | --- | --- | --- | --- | --- |
| | 8k | 16k | 32k | 64k | 128k | LCB | CF | Code Avg | GPQA-D | AIME24 | AIME25 | MATH500 | Math Avg |
| Granite-3.3 Base | 85.81 | 82.40 | 75.53 | 64.91 | 59.09 | 2.15 | 1.99 | 2.07 | **22.56** | 0.46 | 0.31 | 26.09 | 8.95 |
| Mid-Train (Math+Code) | 89.02 | 60.44 | 21.52 | 11.71 | 6.46 | 11.11 | 10.30 | 10.71 | 19.02 | 32.44 | 28.33 | 74.22 | 44.99 |
| Mid-Train + LC (Attention) | 90.04 | 82.56 | 71.47 | 54.63 | 36.32 | 23.78 | 15.53 | 19.65 | 17.85 | 36.56 | 32.55 | 67.20 | **45.44** |
| Mid-Train + LC (Full) | 89.29 | 80.74 | 70.86 | 56.02 | 38.41 | 29.99 | 21.04 | 25.52 | 14.48 | 35.21 | 30.36 | 62.30 | 42.62 |
| Merge (15% Base + 85% Mid-Train) | 89.12 | 69.76 | 32.63 | 15.44 | 11.32 | 10.75 | 10.96 | 10.86 | 22.22 | 28.39 | 24.90 | 72.97 | 42.09 |
| Merge + LC (Attention) | 90.00 | 84.27 | 73.31 | 57.27 | 37.75 | 26.16 | 17.29 | 21.73 | 17.51 | 33.85 | 28.75 | 71.28 | 44.63 |
| Merge + LC (Full) | 89.83 | 84.08 | 73.89 | 60.06 | **42.16** | 29.51 | 21.56 | **25.54** | 15.82 | 33.75 | 30.78 | 68.91 | 44.48 |

*Table 6.* Restoring long-context capability after mid-training for Granite-3.3 (8B). RULER is evaluated from 8k to 128k input lengths. Downstream performance includes Code (LiveCodeBench, Codeforces), Science (GPQA-Diamond), and Math (AIME 24, AIME 25, MATH500).

are conducted using the Granite-4 Micro (3B) dense model to enable efficient ablation.

Relative to the base model, mid-training yields large gains in both math and code with modest budgets. At 10.49B tokens, Math Avg increases from 26.28 to **40.21 (+13.93)**, while Code Avg improves from 1.26 to **9.59**. Increasing the budget to 15.73B further improves Math Avg to **42.07** while maintaining a strong Code Avg of **9.02**.

Beyond 26.21B tokens, gains largely saturate. Math Avg remains nearly constant (42.22 to **42.42**), while Code Avg declines from 8.69 to 7.62 as the budget increases to 31.46B. General-purpose leaderboard scores (LB V1 and V2) remain stable across budgets, and GPQA-Diamond shows no consistent trend. These results indicate that most benefits of Math+Code mid-training are realized within approximately 15B to 27B tokens for this model.

| Budget (B) | LB V1 | LB V2 | Code Avg | GPQA-D | Math Avg |
| --- | --- | --- | --- | --- | --- |
| Base | 66.01 | 21.82 | 1.26 | 21.55 | 26.28 |
| 10.49 | 63.45 | 19.50 | **9.59** | 19.19 | 40.21 |
| 15.73 | 63.24 | 19.79 | 9.02 | **23.06** | 42.07 |
| 26.21 | 63.28 | 19.63 | 8.69 | 19.19 | 42.22 |
| 31.46 | 63.16 | 20.05 | 7.62 | 21.38 | **42.42** |

*Table 7.* Effect of increasing mid-training token budget on Granite-4 Micro (3B) using the Math+Code mixture. Code Avg averages LiveCodeBench and Codeforces; Math Avg averages AIME24, AIME25, and MATH500. Full table is reported in Appendix 9.

## 8. Effects of Reinforcement Learning on Mid-Trained Models

> **Takeaway:** Reinforcement learning (RL) on mid-trained models gives a further boost in scores across all the math (**+5 points**), code (**+8 points**), and science (**+18 points**) reasoning benchmarks. The gains are substantially larger than those obtained by applying RL directly to base models.

In this section, we will highlight the key decisions we took to further improve the performance of different models across math, code, and science reasoning benchmarks. The training

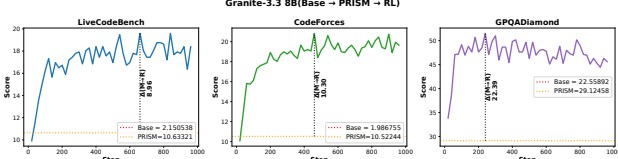

*(a)* Granite-3.3-8B (Code & Science)

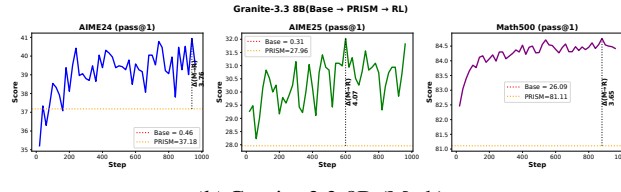

*(b)* Granite-3.3-8B (Math)

*Figure 3.* RL on top of PRISM for Granite-3.3-8B.

hyperparameters we used worked across different model families and resulted in stable performance improvements. Algorithm and training details are provided in Appendix F.

**Data Filtering.** Details about the different datasets used are provided in Appendix D. We prepared two data mixes: one containing math and code domains and the other containing math, code, and science domains. We used the Granite-3.3-8b mid-trained model trained on math, code, and science domains to filter the training prompts. Filtering medium-difficulty prompts using a mid-trained model gave us the best results. More details are provided in Appendix E.

**Results and Findings.** We studied the improvements and trends achieved by doing RL on top of mid-trained models. Details for Mistral-24B and Granite-3.3-8B are provided in Figs. 3 and 4. Details of Mistral-7B, Llama-3.1-8B, and Granite-4-micro-dense are provided in Appendix G.

**Improvements across benchmarks.** RL on top of PRISM yields consistent, positive gains across nearly all benchmarks and model families. GPQADiamond shows the largest absolute improvements in many cases (example: Mistral-24B: **+27.95** absolute increase over PRISM, Granite-3.3 8B: **+22.39** absolute increase over PRISM).

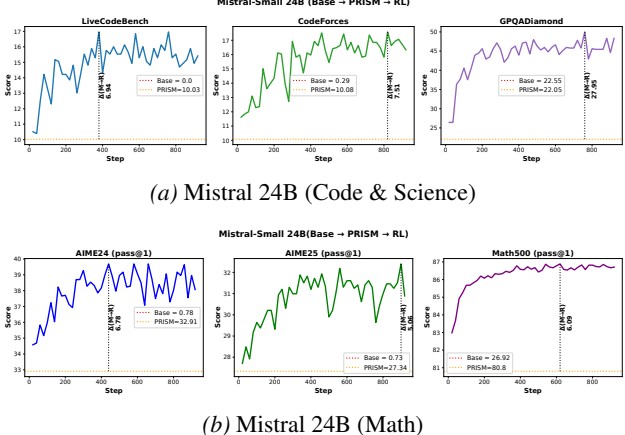

*(a)* Mistral 24B (Code & Science)

*(b)* Mistral 24B (Math)

*Figure 4.* RL on top of PRISM for Mistral 24B.

LiveCodeBench gains are substantial (examples: Granite-3.3 **+8.96**, Mistral-24B **+6.94**), indicating better code generation after PRISM $\rightarrow RL$ (which can also be seen in Appendix J.8). CodeForces improvements are more variable (range observed roughly **+2.65 to +10.30**), with Granite-3.3 showing the biggest CodeForces bump (**+10.30**). Improvements in math benchmarks are smaller in absolute terms than the biggest GPQA gains but still consistent. AIME24/AIME25 gains typically fall in the **3–7** absolute range across models.

**Effect of RL on PRISM vs base models.** We apply RL directly to the Granite 3.3 8B base model to compare both final performance and improvements over multiple training steps across math, code, and science benchmarks. Our results are consistent with prior findings (Wang et al., 2025; Zhang et al., 2025), and highlight that PRISM provides a substantially stronger initialization, enabling more effective expansion of reasoning capabilities through on-policy RL. More details are provided in Appendix H.

**Improvements across model families.** Mistral-24B has the largest absolute gains in GPQADiamond, along with solid gains on CodeForces and LiveCodeBench. Granite-3.3-8B shows the greatest code improvements in absolute terms. Mistral-24B shows non-saturating improvements across all the math, code, and science benchmarks. Other models show similar consistent improvements (for more details, refer to Appendix G). Overall, all the model families benefit from RL on top of PRISM.

**Further boost in numbers.** Across both code and math benchmarks, many RL curves continue to trend upward or exhibit oscillations around an improving mean rather than clean saturation. This is visible in LiveCodeBench, CodeForces, AIME24/25, and Math500, where scores often keep improving late into training. This suggests that the PRISM $\rightarrow RL$ pipeline has not yet exhausted the avail-

able performance gains. Several models show noticeable improvements well after hundreds of RL steps (e.g., Granite-3.3 on CodeForces and LiveCodeBench; Mistral-24B on Codeforces and Math500). This indicates that longer RL schedules or additional RL phases could plausibly yield further gains, especially when starting from a strong mid-trained (PRISM) initialization. This strengthens the case for viewing PRISM not as a final training stage, but as a launch point for deeper RL or multi-stage RL pipelines.

**RL on hybrid models.** We trained the PRISM-initialized Granite-4-H Micro hybrid model using the same RL recipe applied to other model families. While the model exhibited initial improvements, training became unstable and collapsed after a few hundred optimization steps, resulting in sub-optimal final performance. We explored multiple RL data mixtures and hyperparameter settings in an effort to improve stability; these investigations are ongoing. As future work, we plan to further reduce training–inference mismatch (Team et al., 2025b; Yao et al., 2025) and incorporate stabilization techniques aimed at maintaining policy entropy throughout RL training (Shen et al., 2025; MiniMax et al., 2025). Appendix I presents representative examples of the training instabilities observed.

## 9. Conclusion and Future Work

We presented PRISM, a retention-aware framework for systematically studying mid-training in large language models. Across model families, data mixtures, and benchmarks, we showed that mid-training performance is strongly governed by data composition. Balanced mixtures that combine general web and instruction data with domain-specific reasoning sources yield consistent gains in math, code, and science while preserving general and long-context capabilities. We further showed that mid-training provides a stable foundation for downstream reinforcement learning, enabling reliable improvements across reasoning tasks.

As future work, we plan to extend PRISM to additional domains and study reinforcement learning on hybrid architectures. While PRISM-mid-trained dense models support stable RL, we observe that RL on top of mid-trained hybrid models remains challenging, highlighting an important direction for further investigation.

## Impact Statement

This paper studies mid-training strategies for large language models with the goal of improving reasoning performance while preserving general capabilities. Our work is methodological and empirical in nature, focusing on understanding training design choices rather than deploying new applications. While improvements in model reasoning may enable more capable downstream applications, we do not antic-

ipate immediate negative societal impacts beyond those commonly associated with large language models, such as misuse or bias inherited from training data. Addressing these broader concerns remains an important area of ongoing research.

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

| Model / Mixture | Code | | | Science | Math | | | |
|---|---|---|---|---|---|---|---|---|
| | LiveCodeBench | Codeforces | Code Avg | GPQA-Diamond | AIME 24 | AIME 25 | MATH 500 | Math Avg |
| Base | 2.15 | 1.99 | 2.07 | 22.56 | 0.46 | 0.31 | 26.09 | 8.95 |
| Math only | 2.15 | 3.46 | 2.81 | 17.34 | 26.72 | 22.08 | 60.50 | 36.43 |
| Math + Code | 11.11 | 10.30 | **10.71** | 19.02 | 32.44 | 28.33 | 74.22 | 44.33 |
| Math + Code + Science | 10.63 | 10.52 | *10.58* | **29.12** | 37.18 | 27.96 | 81.11 | **48.75** |

*Table 8.* Domain-specific evaluation results for Granite-3.3-8B. Code Avg is the mean of LiveCodeBench and Codeforces; Math Avg is the mean of AIME 24, AIME 25, and MATH 500.

| Token Budget (B) | LB V1 | LB V2 | LCB | CF | Code Avg | GPQA-D | AIME24 | AIME25 | MATH500 | Math Avg |
|---|---|---|---|---|---|---|---|---|---|---|
| Base | 66.01 | 21.82 | 0.24 | 2.28 | **1.26** | 21.55 | 16.09 | 12.34 | 50.42 | **26.28** |
| 10.49 | 63.45 | 19.50 | 10.51 | 8.68 | **9.59** | 19.19 | 23.95 | 19.17 | 77.52 | **40.21** |
| 15.73 | 63.24 | 19.79 | 10.75 | 7.28 | **9.02** | 23.06 | 26.14 | 21.30 | 78.76 | **42.07** |
| 26.21 | 63.28 | 19.63 | 9.80 | 7.58 | **8.69** | 19.19 | 28.49 | 20.10 | 78.08 | **42.22** |
| 31.46 | 63.16 | 20.05 | 8.24 | 6.99 | **7.62** | 21.38 | 28.02 | 22.08 | 77.15 | **42.42** |

*Table 9.* Effect of increasing mid-training token budget on Granite-4 Micro (3B) using the Math+Code mixture with fixed 8k context length. LCB denotes LiveCodeBench, CF denotes Codeforces, and GPQA-D denotes GPQA-Diamond. **Code Avg** averages LCB and CF, while **Math Avg** averages AIME24, AIME25, and MATH500. All values are reported to two decimal places.

# A. Dataset Details

## A.1. Dataset Preprocessing Details

This section describes the dataset-specific preprocessing steps used to construct the mid-training mixtures.

**Web data filtering.** For general web data, we use the DCLM-EDU corpus and retain documents with a quality score greater than or equal to 3, following the dataset's recommended filtering guidelines. This step removes low-quality or noisy documents while preserving broad coverage of general knowledge.

**Reasoning datasets.** For OpenCodeReasoning-2, we retain only samples whose judgment is marked as `right` by the QwQ evaluator model and for which sufficient test coverage is available (i.e., `pass_rate` $\neq -1$). From this filtered pool, we randomly sample 60k Python examples and 60k C++ examples. Other reasoning datasets are used as provided, without additional filtering beyond standard deduplication.

**Chat and instruction-following data.** For chat-style datasets, all conversations are normalized by explicitly prefixing utterances with speaker roles ("User:" and "Assistant:"). For WildChat-1M, we further restrict the data to high-quality conversations generated by GPT-4, following prior evidence that such filtering improves stability in downstream reinforcement learning.

For all reasoning datasets and chat data, we concatenate the question and answer with a single line break between them, following the work of (Wang et al., 2025).

# B. Model and Training Details

## B.1. Model Details and Specifications

To evaluate the cross-architecture robustness of PRISM, we select a diverse set of Large Language Models (LLMs) ranging from 3B to 24B parameters. These include traditional dense Transformers, State-Space Model (SSM) hybrids, and Mixture-of-Experts (MoE) designs.

**Dense Transformer Models:** We utilize the **LLaMA-3.1 8B** (Grattafiori et al., 2024) and **Mistral-7B-v0.1** as primary baselines. For enterprise-focused dense evaluation, we include **Granite-3.3 8B** and the lightweight **Granite-4.0 Micro 3B**, alongside the larger **Mistral-Small-24B**.
**Hybrid Attention-Mamba (SSM) Architectures:** We include **Granite-4.0-H Micro 3B**, which represents a hybrid paradigm. This architecture alternates between standard *Attention* layers and *Mamba2* layers.
**Hybrid MoE Architectures:** We also evaluate **Granite-4.0-H Tiny 7B**. This model combines the Attention-Mamba hybrid

| Category | Setting |
|---|---|
| Training steps | 25,000 |
| Micro batch size | 1 |
| Gradient accumulation steps | 1 |
| Effective batch size | 1 |
| Optimizer | AdamW |
| Learning rate | $5 \times 10^{-5}$ |
| Weight decay | 0.1 |
| Adam $\beta_1, \beta_2$ | (0.9, 0.95) |
| Adam $\epsilon$ | $1 \times 10^{-10}$ |
| Learning rate schedule | Cosine decay |
| Warmup steps | 500 |
| Decay steps | 24,500 |
| Final LR factor | 0.1 |
| Precision | bfloat16 (bf16) |
| FSDP algorithm | 2 |
| Data parallel sharding | 8 |
| Data parallel replication | 16 |

*Table 10.* Training hyperparameters used for PRISM mid-training.

| Category | Setting |
|---|---|
| Training steps | 1,000 |
| Micro batch size | 1 |
| Gradient accumulation steps | 1 |
| Effective batch size | 1 |
| Evaluation during training | Disabled |
| Evaluation interval | $10^9$ steps |
| Optimizer | AdamW |
| Learning rate | $5 \times 10^{-5}$ |
| Weight decay | 0.1 |
| Adam $\beta_1, \beta_2$ | (0.9, 0.95) |
| Adam $\epsilon$ | $1 \times 10^{-10}$ |
| Learning rate schedule | Exponential decay |
| Warmup steps | 100 |
| Constant steps | 0 |
| Final LR factor | 0 |
| Precision | bfloat16 (bf16) |
| FSDP algorithm | 2 |
| Context parallelism | 4 |
| Data parallel sharding | 4 |
| Data parallel replication | 9 |
| Gradient checkpointing | Enabled |

*Table 11.* Training hyperparameters for long-context restoration after mid-training.

| Dataset | Type | Tokens (B) |
|---|---|---|
| DCLM-EDU | General Web Data | 2.51 |
| Nemotron Post-Training v1 | Math (QA/Reasoning) | 5.08 |
| Megamath-Web-Pro | Math (web) | 4.33 |
| StarCoder2 | Code (web) | 37.52 |
| xenArcAI-codex | Code (QA/Reasoning) | 3.860 |

*Table 12.* Datasets used in Long-Context Extension Phase. Token counts are reported in billions using Granite 3.3 (8B).

block with a *Mixture-of-Experts (MoE)* modules.

### B.2. PRISM Training Details

Table 10 summarizes the training hyperparameters used for PRISM mid-training across all models unless otherwise specified.

### B.3. Long Context Extension Phase

Table 11 summarizes the hyperparameters used for the long-context extension phase applied after mid-training and Table 12 list the datasets used for long-context phase.

**Long-context sequence packing via Best-Fit Decreasing (BFD).** To efficiently construct fixed-length long-context training sequences while minimizing truncation and wasted capacity, we employ a Best-Fit Decreasing (BFD) packing strategy. Given a collection of tokenized documents with variable lengths, documents are first optionally split into overlapping chunks if they exceed the target context length $L$. All resulting chunks are then sorted in decreasing order of length. Each chunk is greedily assigned to an existing sequence buffer whose remaining capacity is sufficient and minimal among all feasible buffers; if no such buffer exists, a new buffer is created. This process continues until all chunks are assigned. The resulting buffers correspond to packed training sequences of length at most $L$, which are emitted without further splitting except for a final truncation to exactly $L$ tokens if necessary. Compared to naive concatenation or first-fit strategies, BFD packing

| Context | LB V1 | LB V2 | LCB | CF | GPQA-D | AIME24 | AIME25 | MATH500 |
|---|---|---|---|---|---|---|---|---|
| Base | 66.01 | 21.82 | 0.24 | 2.28 | 21.55 | 16.09 | 12.34 | 50.42 |
| PRISM 8k | 62.91 | 20.53 | 10.87 | 6.25 | 34.34 | 27.19 | 22.29 | 79.40 |
| PRISM 16k | 64.23 | 20.37 | 12.19 | 8.90 | 38.89 | 31.82 | 25.26 | 82.47 |
| PRISM 32k | 64.41 | 20.35 | 12.54 | 9.79 | 24.24 | 32.08 | 20.72 | 81.34 |

*Table 13.* Effect of mid-training context length on Granite-4 Micro (3B) using the Math+Code+Science mixture. LCB denotes LiveCodeBench, CF denotes Codeforces, and GPQA-D denotes GPQA-Diamond. All values are reported to two decimal places.

significantly reduces unnecessary document truncation and improves token utilization while preserving document-level coherence within long contexts.

### B.4. Effect of Mid-Training Context Length

In this section we study the effect of increasing the mid-training context length while keeping the data mixture fixed to Math+Code+Science and maintaining a comparable token budget. To enable rapid iteration, all ablations are conducted using the Granite-4 Micro (3B) dense model. We compare PRISM mid-training with context lengths of 8k, 16k, and 32k tokens against the base model, and evaluate performance on general-purpose leaderboards as well as domain-specific benchmarks for math, code, and science.

Increasing the mid-training context length leads to consistent improvements in domain-specific reasoning relative to the base model. With the Math+Code+Science mixture, PRISM mid-training at 8k context raises MATH 500 from 50.42 to **79.40** (+28.98 points) and improves LiveCodeBench from 0.24 to **10.87**. Expanding the context to 16k yields further gains across domains: MATH 500 increases to **82.47**, AIME 24 improves from 27.19 to **31.82**, Codeforces rises from 6.25 to **8.90**, and GPQA-Diamond improves from 34.34 to **38.89**. These results indicate that moderate long-context mid-training strengthens the model's ability to leverage multi-step reasoning signals present in math, code, and science data.

However, gains largely saturate beyond 16k context. While increasing context to 32k yields small additional improvements on some benchmarks, such as Codeforces (8.90 to **9.79**) and AIME 24 (31.82 to **32.08**), other metrics plateau or regress slightly, including MATH 500 (82.47 to 81.34) and GPQA-Diamond (38.89 to 24.24). General-purpose performance remains relatively stable across context lengths, with Leaderboard V1 partially recovering from 62.91 at 8k to **64.41** at 32k and Leaderboard V2 staying near 20.4 throughout. Overall, these trends suggest that increasing mid-training context length is beneficial up to a point, with 16k providing the most favorable balance between reasoning gains and training efficiency for this model and data mixture.

## C. Evaluation Details

**Benchmark Details.** To evaluate the effectiveness of mid-training design choices, a comprehensive suite of benchmarks is required to capture distinct dimensions. General ability is often assessed via LB-V1 (ARC, HellaSwag, MMLU, TruthfulQA, WinoGrande, and GSM8K) and LB-V2 (utilizing more rigorous tasks like IFEval, BBH, MATH, GPQA, MuSR, and MMLU-Pro), which serve to detect generalization regressions and ensure the model retains broad multitask knowledge. Long-context capabilities are specifically validated by RULER (Hsieh et al., 2024), which moves beyond simple retrieval to measure "effective" reasoning across massive token windows. In the domain of code, LiveCodeBench and provide a contamination-free signal by using time-stratified problems and elite algorithmic challenges to distinguish genuine program synthesis from memorized training data. Mathematical proficiency is tracked through and MATH500, benchmarks highly sensitive to data mixture quality and symbolic reasoning gains. Finally, for high-tier scientific reasoning, GPQA-Diamond offers "Google-proof" challenges that probe PhD-level expertise, ensuring the model can handle complex deduction that exceeds surface-level fact recall.

**Evaluation Settings.** For Math benchmarks MATH500, AIME24/25 we use 32k max generation tokens with temperature of 0.6 and top-p 0.95, generate 64 samples per prompt.

For Code Benchmarks (Codeforces, LiveCodeBench) we use 32k max generation tokens with temperature 0.7, and generate 3 samples per prompt.

| Domain | Sources | Count |
|---|---|---|
| Math Prompts | DeepScaleR-Preview
INTELLECT-2-RL
Skywork-OR1-RL-Data-math | 294K |
| Science Prompts | Nemotron-Post-Training-Dataset-v1-stem | 100K |
| Code Prompts | DeepCoder-Preview
Skywork-OR1-RL-Data-code
OpenCodeInstruct | 142K |

*Table 14.* Datasets used in RL mixtures across Math, Science, and Code domains, along with prompt counts per domain.

## D. RL Data

Table 14 shows the different datasets used to form the mixtures for math, code, and science domains, along with their respective counts.

## E. Data Filtering

We prepared two data mixes: one containing math and code domains and the other containing math, code, and science domains using the Granite-3.3-8b mid-trained model to filter the training prompts. For each prompt, we sampled 16 responses, using a temperature of 1.0 and a top_p of 1.0. For the math domain, we initially selected prompts having $\geq 1$ and $\leq 8$ correct samples (out of 16), which yielded a large pool of candidates. To construct a harder subset, we further restricted this set to prompts with exactly one correct sample; this criterion proved effective across different model families. In contrast, for the code and science domains, most RL prompts were unsolvable. We therefore retained all prompts with at least one correct sample (out of 16). This procedure resulted in 19k math prompts, 17k science prompts, and 7k code prompts. Despite the imbalance in different domains, we got good improvements across all reasoning benchmarks.

## F. RL Training details

$$\mathcal{J}_{GRPO}(\theta) = \mathbb{E}_{q \sim P(Q), \{o_i\}_{i=1}^{G} \sim \pi_{\theta_{gen}}} \left[ \frac{1}{\sum_{j=1}^{G} |o_j|} \sum_{i=1}^{G} \sum_{t=1}^{|o_i|} \right.$$

$$\min \left( \frac{\pi_{\theta_{old}}(o_{i,t}|q)}{\pi_{\theta_{gen}}(o_{i,t}|q)}, C \right) \times$$

$$\min \left( \frac{\pi_\theta(o_{i,t}|q)}{\pi_{\theta_{old}}(o_{i,t}|q)} \hat{A}_i, \right.$$

$$\left. \text{clip} \left( \frac{\pi_\theta(o_{i,t}|q)}{\pi_{\theta_{old}}(o_{i,t}|q)}, 1 - \epsilon_{\text{low}}, 1 + \epsilon_{\text{high}} \right) \hat{A}_i \right)$$

$$\left. - \beta \mathbb{D}_{KL}[\pi_\theta || \pi_{ref}] \right] \quad (1)$$

$$\hat{A}_i = \frac{r_i - \text{mean}(r_1, \ldots, r_G)}{\text{std}(r_1, \ldots, r_G) + \eta} \quad (2)$$

We used Group Relative Policy Optimization (Shao et al., 2024) as our RL algorithm for all the models. The equations for the same are provided in 1, 2. RL training configuration is provided in Table 15.

We use on-policy model updates for all our experiments, i.e, our $train\_batch\_size = inference\_batch\_size$. Following (Gorbatovski et al., 2025), we soft-update the reference policy for better convergence. Since our mid-training data teaches the model to think, we apply a *thinking format reward* during RL training. Specifically, the model's response is evaluated for correctness only if it adheres to the format: `<think> thoughts </think> response`. Otherwise, it is assigned a

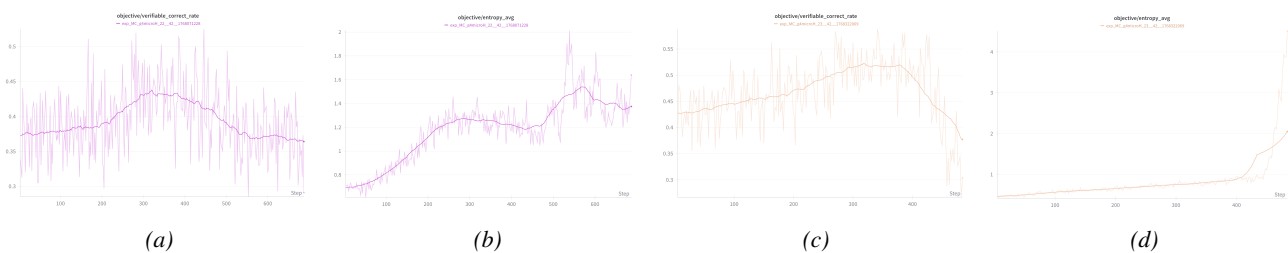

*Figure 5.* Examples of RL training instability in the hybrid model.

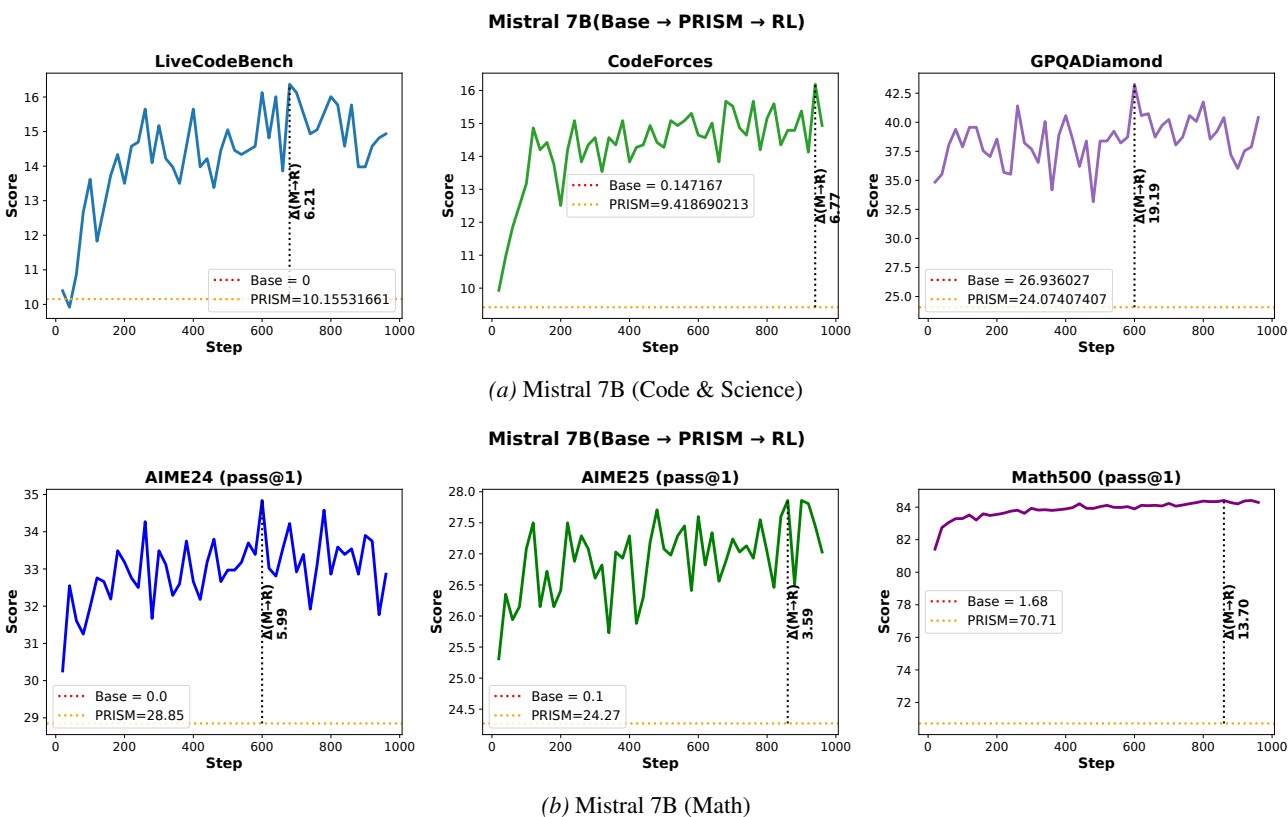

*Figure 6.* RL on top of PRISM for Mistral 7B.

reward of 0.0. We also apply a stop token penalty to teach the model to end its answers well and a repetition penalty to control repetitions in model reasoning. Our models were trained using on-policy updates, and thus, the importance sampling ratio between the current policy and the old policy is one. We also applied the Truncated Importance Sampling ratio (Yao et al., 2025) to control for training-inference mismatch in all our training runs. We train our models using GB200 nodes and use the open-instruct codebase[1] for all our experiments.

# G. RL Training Improvements

Figures 6, 7, 8 show the improvements in benchmark performance for Mistral-7B, Llama-3.1-8B, and Granite-4-micro-dense. Strong GPQA gains are observed across Mistral-7B: **+19.19**, Llama: **+18.35**, and Granite-4-micro: **+7.91**. LiveCodeBench gains are substantial (examples: Llama **+8.96**, Granite-4-micro **+5.62**, Mistral-7B **+6.21**), indicating improved code generation capabilities. Granite-4 Micro (3B) has consistent improvements across benchmarks but is generally smaller in absolute terms compared with larger 8B models. Overall, all the model families show consistent improvements in

---

[1]https://github.com/allenai/open-instruct/tree/main

*Table 15.* RL training hyperparameters and training configuration

| Hyperparameter | Value |
|---|---|
| **Sampling Parameters** | |
| Responses per Prompt | 16 |
| Temperature | 1.0 |
| Top-$p$ | 1.0 |
| **Training Dynamics** | |
| Total Batch Size | 1024 |
| Unique Prompts per Batch | 64 |
| Training Steps | 1,000 |
| Context Length | 16,384 |
| Learning Rate | $5 \times 10^{-7}$ |
| LR Schedule | Linear Decay |
| KL-loss Beta ($\beta$) | 0.05 |
| **RL Environment & Rewards** | |
| Importance Sampling Ratio | 1.0 (On-policy) |
| Truncated IS Constant ($C$) | 2.0 |
| Format Reward | `<think> ... </think>` |
| Penalties | Stop token, Repetition |
| Reference Policy Update | Soft-update |
| **Infrastructure** | |
| Optimizer | AdamW |
| Parallelism Strategy | DeepSpeed Stage 3 |
| Packing Length | 16,384 |
| Hardware | GB200 Nodes |

benchmarks owing to RL on top of PRISM.

### G.1. Overall RL improvements

We show the results of the RL-trained models in Table 16. We used two variations of the mid-trained model for RL: one trained on the math and code mix and the second trained on the math, code, and science mix. For both these variants, we did RL on both math and code and math, code, and science mixes, resulting in 4 RL-trained models for each model family.

**Step 1: Base → Midtraining.** Across all three model families, midtraining (MC or MCS) produces the largest performance jump over the base models. For LLaMA, AVG increases from 7.49 (base) to 21.33 with MC midtraining (+13.84). For g33, AVG improves from 11.19 to 24.69 with MC (+13.50) and further to 29.48 with MCS (+18.29). For Mistral, AVG rises from 9.20 to 18.49 with MC (+9.29) and to 25.05 with MCS (+15.85). These gains are driven primarily by large improvements on math and code benchmarks (MATH500, MATH_AVG, and CODE_AVG), showing that domain-specific midtraining is critical for downstream reasoning performance.

**Step 2: Impact of Midtraining Data Mix.** Comparing MC and MCS midtraining reveals that adding science data (MCS) has a model-dependent effect. LLaMA shows minimal sensitivity to the midtraining mix (21.33 vs. 21.07 AVG), whereas g33 and Mistral benefit substantially from MCS, gaining +4.79 and +6.56 AVG points respectively over MC. This indicates that some architectures can better leverage cross-domain supervision during midtraining.

**Step 3: Midtraining → RL Fine-tuning.** Applying RL on top of midtrained checkpoints consistently improves performance further. With MC midtraining, RL increases AVG by +4.57 for LLaMA (21.33 → 25.90), +11.47 for g33 (24.69 → 36.16), and +12.99 for Mistral (18.49 → 31.48). Similar but slightly smaller gains are observed when starting from MCS midtraining. RL improvements are particularly strong on GPQA and CODE_AVG, indicating better instruction-following and answer calibration beyond the gains achieved by midtraining alone.

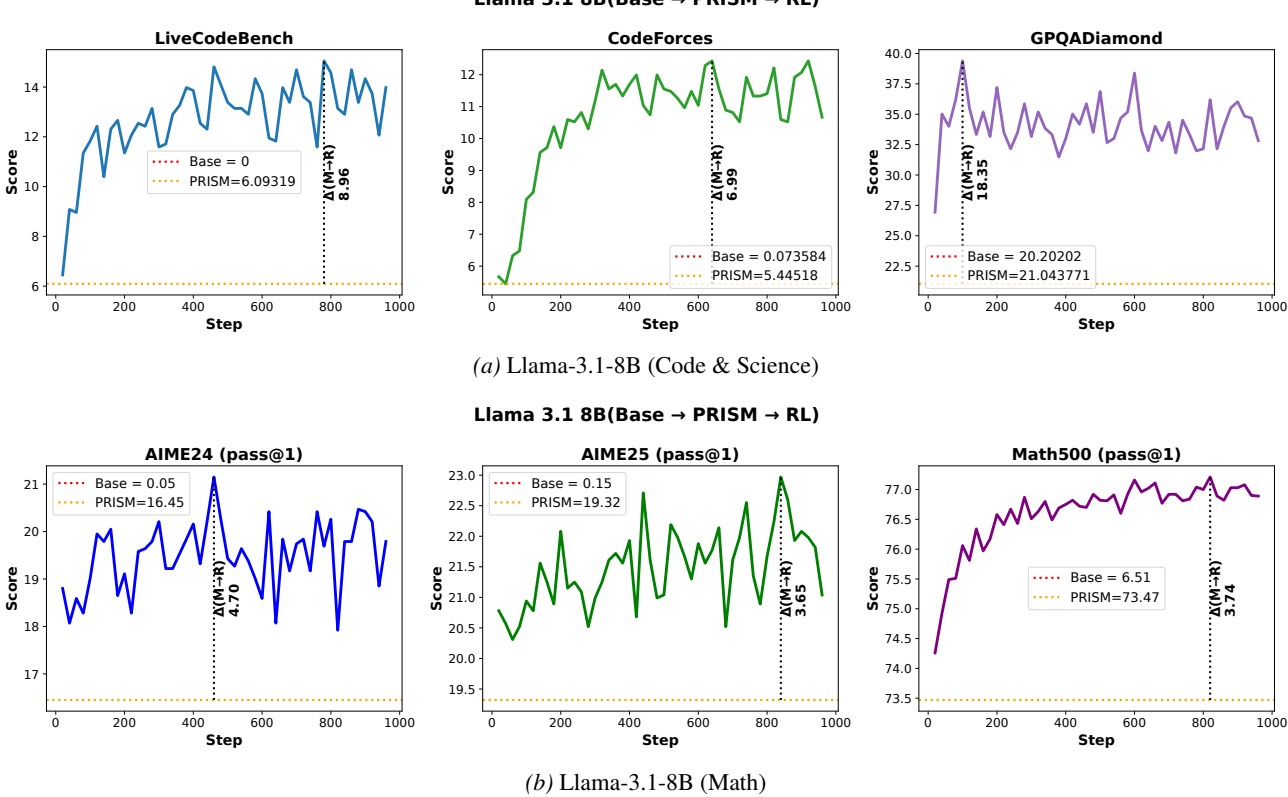

*(a)* Llama-3.1-8B (Code & Science)

*(b)* Llama-3.1-8B (Math)

*Figure 7.* RL on top of PRISM for Llama-3.1-8B.

**Step 4: Impact of RL Data Mix.** Changing the RL mix from MC to MCS results in comparatively small and model-dependent differences (typically $< 2$ AVG points). In several cases, matching the RL mix to the midtraining mix yields marginal gains, but the overall effect of the RL mix is significantly smaller than that of the midtraining mix.

**Step 5: Overall Trend.** Overall, the data shows a clear hierarchy of effects: base $\rightarrow$ midtraining provides the dominant gains, RL adds a consistent second-stage improvement, and data mix choices matter most during midtraining. For math- and code-centric evaluations, MC midtraining followed by RL yields strong performance, while incorporating science data at midtraining (MCS) can further benefit models that effectively utilize cross-domain signals.

## H. Effect of RL on Base models

Figure 9 presents learning curves for the reasoning benchmarks at different stages of training. While RL leads to noticeable gains on Math500, coding, and science tasks, the model fails to consistently improve over the base checkpoint on AIME24 and AIME25. Overall, RL on the base model underperforms RL on PRISM by a large margin, with final scores lower by **39.66 points** in math, **14.01 points** in code, and **21.72 points** in science.

## I. RL Training instabilities in the hybrid model

Figure 5 illustrates the RL training instabilities observed in two experiments with the Granite-4H-Micro model. Figures 5a and 5b correspond to experiment exp_22, which uses a temperature of 1.0, less frequent reference policy updates, and otherwise identical hyperparameters to the earlier configuration. In this setting, the correctness rate initially improves but subsequently degrades, while the policy entropy steadily increases, ultimately leading to training instability. During this phase, we also observed the generation of incoherent outputs for incorrect samples. We hypothesize that large negative gradients associated with such samples may contribute to the observed instability.

Figures 5c and 5d present results obtained with a reduced temperature of 0.8 while keeping all other hyperparameters

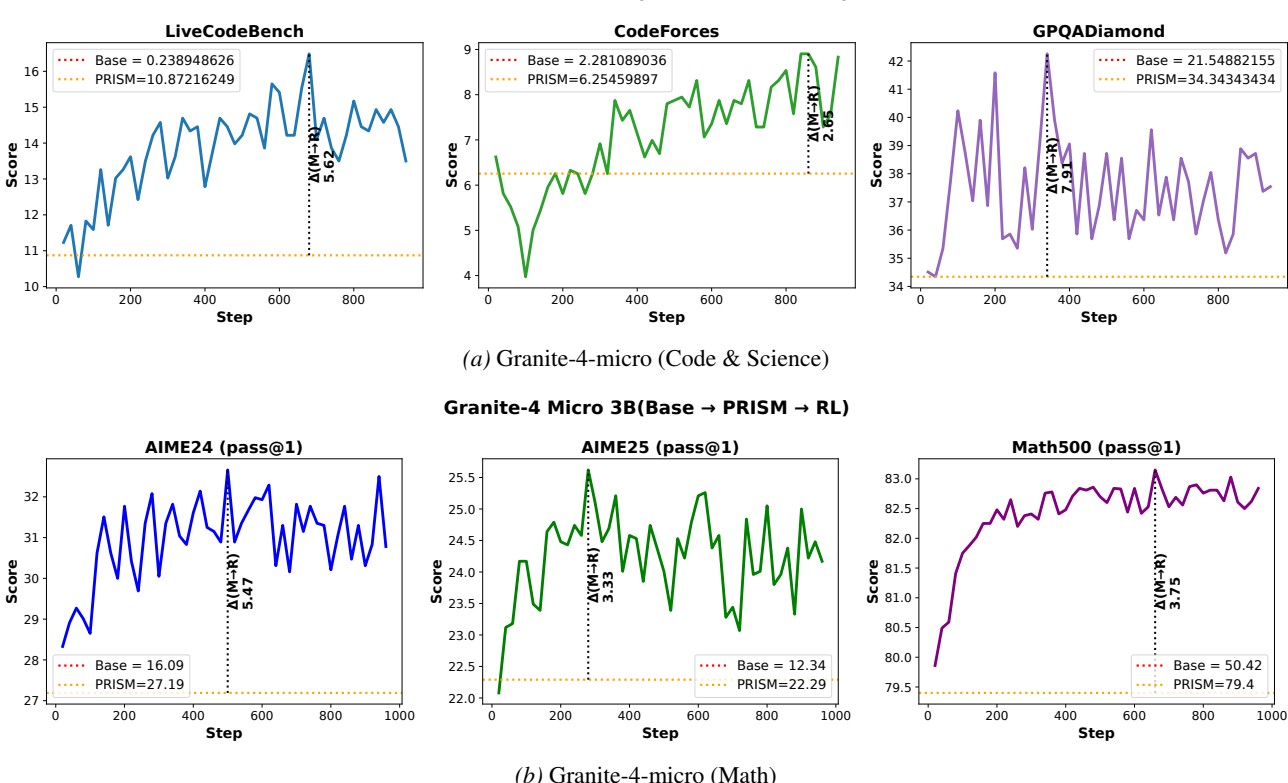

*(a)* Granite-4-micro (Code & Science)

*(b)* Granite-4-micro (Math)

*Figure 8.* RL on top of PRISM for Granite-4-micro dense model.

identical to exp_22. Although the correctness rate again improves in the early stages of training, it eventually collapses in a manner similar to the higher-temperature setting. Notably, the entropy exhibits a monotonic increase from the beginning of training, indicating a risk of model collapse if left uncontrolled. We plan to further investigate the underlying causes of these instabilities and report our findings in future work.

## J. Model Generations

Here, we have shown the Granite-3.3-8b model generations for math, code, and science domains at different stages of training.

### J.1. Math Prompt

```
In a car dealership with some cars, 60% of the cars are hybrids, and 40% of the
hybrids contain only one headlight. There are 216 hybrids with full headlights. How
many cars are there in the dealership?

Please put your final answer in \\boxed{} format.
```

### J.2. Granite 3.3 8b base math generation

The correct answer for the above problem is 600; however, the base model gets it wrong probably because it rushes to the final answer (using **192 tokens**) without reasoning about the possible steps.

| Experiment | MT | RL | LCB | CF | CODE_AVG | AIME24 | AIME25 | MATH500 | MATH_AVG | GPQA | AVG |
|---|---|---|---|---|---|---|---|---|---|---|---|
| llama_base | - | - | 0.00 | 0.07 | 0.04 | 0.05 | 0.15 | 6.51 | 2.24 | 20.20 | 7.49 |
| g33_base | - | - | 2.15 | 1.99 | 2.07 | 0.46 | 0.31 | 26.09 | 8.95 | 22.56 | 11.19 |
| mistral_base | - | - | 0.00 | 0.15 | 0.07 | 0.00 | 0.10 | 1.68 | 0.59 | 26.94 | 9.2 |
| llama_MC | MC | - | 6.93 | 6.03 | 6.48 | 20.67 | 19.58 | 73.7 | 37.98 | 19.53 | 21.33 |
| llama_MCS | MCS | - | 6.09 | 5.45 | 5.77 | 16.45 | 19.32 | 73.47 | 36.41 | 21.04 | 21.07 |
| g33_MC | MC | - | 11.11 | 10.30 | 10.71 | 32.44 | 28.33 | 74.22 | 44.33 | 19.02 | 24.69 |
| g33_MCS | MCS | - | 10.63 | 10.52 | 10.58 | 37.18 | 27.96 | 81.11 | 48.75 | 29.12 | 29.48 |
| mistral_MC | MC | - | 11.11 | 9.27 | 10.19 | 24.63 | 15.52 | 47.7 | 29.28 | 15.99 | 18.49 |
| mistral_MCS | MCS | - | 10.16 | 9.42 | 9.79 | 28.85 | 24.27 | 70.71 | 41.28 | 24.07 | 25.05 |
| MC_llama | MC | MC | 12.31 | 11.85 | 12.08 | 25.47 | 23.23 | 78.99 | 42.56 | 23.06 | 25.9 |
| MCS_llama | MC | MCS | 11.83 | 12.80 | 12.32 | 24.43 | 23.12 | 78.62 | 42.06 | 24.75 | 26.38 |
| MC_llama_MCS | MCS | MC | 13.62 | 11.41 | 12.51 | 20.47 | 21.67 | 77.10 | 39.75 | 34.01 | 28.76 |
| MCS_llama_MCS | MCS | MCS | 14.34 | 12.07 | 13.20 | 20.42 | 22.08 | 77.03 | 39.84 | 36.03 | 29.69 |
| MC_g33 | MC | MC | 20.79 | 18.76 | 19.78 | 40.36 | 33.33 | 85.88 | 53.19 | 35.52 | 36.16 |
| MCS_g33 | MC | MCS | 20.43 | 19.57 | 20.00 | 40.10 | 30.89 | 85.51 | 52.17 | 35.69 | 35.95 |
| MC_g33_MCS | MCS | MC | 20.31 | 20.46 | 20.38 | 40.62 | 30.89 | 84.62 | 52.04 | 52.86 | 41.76 |
| MCS_g33_MCS | MCS | MCS | 17.20 | 18.03 | 17.62 | 40.42 | 29.58 | 83.99 | 51.33 | 51.52 | 40.16 |
| MC_mistral | MC | MC | 17.08 | 16.34 | 16.71 | 34.11 | 27.50 | 84.18 | 48.60 | 29.12 | 31.48 |
| MCS_mistral | MC | MCS | 16.61 | 15.60 | 16.10 | 33.02 | 26.93 | 83.80 | 47.92 | 28.28 | 30.77 |
| MC_mistral_MCS | MCS | MC | 16.61 | 15.31 | 15.96 | 33.75 | 26.93 | 84.15 | 48.28 | 40.91 | 35.05 |
| MCS_mistral_MCS | MCS | MCS | 16.01 | 15.16 | 15.58 | 32.86 | 27.03 | 84.37 | 48.09 | 41.75 | 35.14 |

*Table 16.* Mid-train results across MATH, CODE, and SCIENCE benchmarks for LLaMA, g33, and Mistral variants. All scores are reported to two decimal places. MC corresponds to math and code mix, MCS corresponds to math,code, and science mix. Column MT corresponds to midtraining mix, RL corresponds to RL mix.

```
Let's solve this problem using Python's sympy library.
```python
import sympy as sp

# let's denote the number of cars as x
x = sp.symbols('x')

# 60% of the cars are hybrids
hybrids = 0.6 * x

# 40% of the hybrids contain only one headlight
hybrids_with_one_headlight = 0.4 * hybrids

# there are 216 hybrids with full headlights
eq = sp.Eq(hybrids - hybrids_with_one_headlight, 216)

# solving for x to get the number of cars
sp.solve(eq, x)[0]
```
Thus there are \boxed{360} cars in the dealership.
```

### J.3. Granite 3.3 8b midtrain math generation

Owing to mid-training, the base model learns to reason before coming to a final answer, and it gives the right answer using a total of **933 tokens**.

```
<think>

Okay, let's see. I need to find the total number of cars in the dealership. The
problem says that 60% of the cars are hybrids, and 40% of the hybrids have only one
```

**Granite-3.3 8B(Base → RL)**

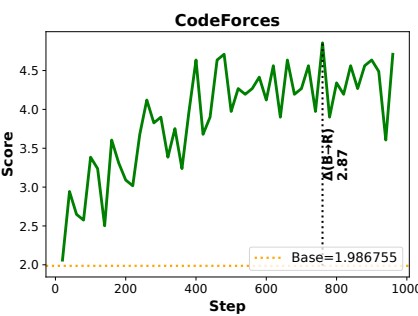
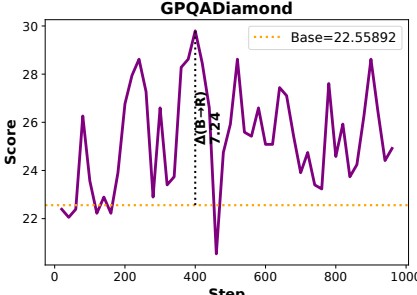

*(a)* Granite 3.3 8B Base (Code & Science)

**Granite-3.3 8B(Base → RL)**

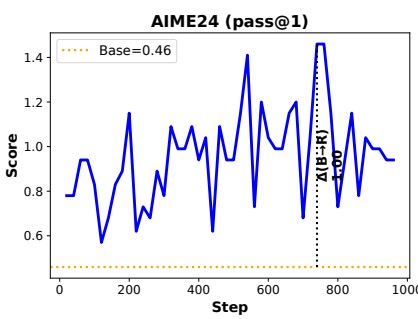
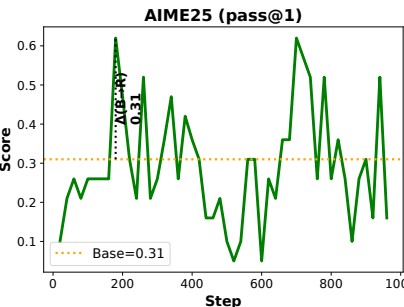
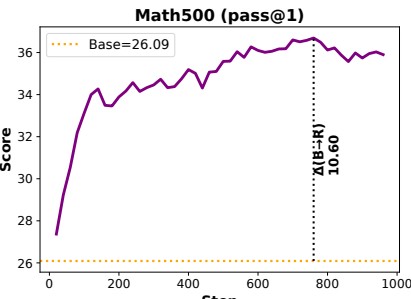

*(b)* Granite 3.3 8B Base (Math)

*Figure 9.* RL on top of Granite 3.3 8B base.

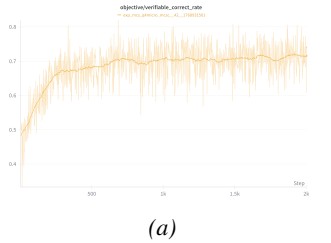
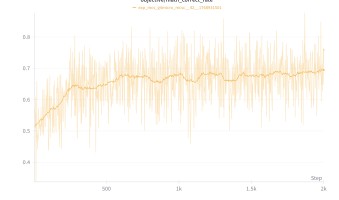
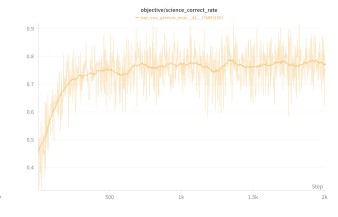
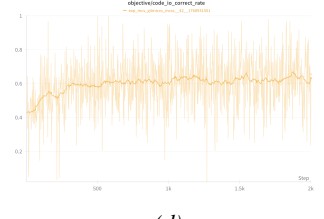

*(a)*        *(b)*        *(c)*        *(d)*

*Figure 10.* RL training correctness curves for G4 micro-dense mictrained model.

```
headlight. There are 216 hybrids with full headlights. Hmm, so I need to relate these
percentages to the total number of cars.

First, let me break down the information. Let's denote the total number of cars as T.
Then, the number of hybrids would be 60% of T, which is 0.6T. Out of these hybrids,
40% have only one headlight. So, the number of hybrids with one headlight is 40% of
0.6T. Let me write that out:

Number of hybrids with one headlight = 40% of hybrids = 0.4 * 0.6T = 0.24T.

But wait, the problem states that there are 216 hybrids with full headlights. So, the
number of hybrids with full headlights is 100% - 40% = 60% of the hybrids. That would
be 0.6 * 0.6T = 0.36T. But they told us that this number is 216. So, 0.36T = 216.

To find T, I can set up the equation:
```

```
0.36T = 216

Then, divide both sides by 0.36:

T = 216 / 0.36

Let me compute that. 216 divided by 0.36. Hmm, 0.36 goes into 216 how many times?

Well, 0.36 * 600 = 216, because 0.36 * 100 = 36, so 36 * 6 = 216. So, 0.36 * 600 = 216.
 Therefore, T = 600.

Wait, so the total number of cars is 600? Let me check that again to make sure I didn'
t make a mistake.

Total cars: 600.

Number of hybrids: 60% of 600 = 0.6 * 600 = 360.

Number of hybrids with one headlight: 40% of 360 = 0.4 * 360 = 144.

Therefore, number of hybrids with full headlights: 360 - 144 = 216. Which matches the
given information. So that checks out.

Therefore, the total number of cars in the dealership is 600.

**Final Answer**
The total number of cars in the dealership is \boxed{600}.
</think>

Let \( T \) be the total number of cars in the dealership.

60% of the cars are hybrids, so the number of hybrids is \( 0.6T \).

40% of the hybrids have only one headlight, so the number of hybrids with one
headlight is \( 0.4 \times 0.6T = 0.24T \).

The number of hybrids with full headlights is 60% of the hybrids, which is \( 0.6 \
times 0.6T = 0.36T \).

We are given that there are 216 hybrids with full headlights. Therefore, we set up the
 equation:
\[
0.36T = 216
\]

Solving for \( T \):
\[
T = \frac{216}{0.36} = 600
\]

Thus, the total number of cars in the dealership is \(\boxed{600}\).
```

### J.4. Granite 3.3 8b $PRISM->RL$ math generation

The RL-trained model gives the right answer too using a total of **917 tokens.**

```
<think>
```

Okay, let's see. So the problem is about a car dealership where 60% of the cars are hybrids. Then, 40% of those hybrids have only one headlight. And we know there are 216 hybrids with full headlights. The question is asking how many cars are in the dealership in total. Hmm, okay.

First, let me parse this step by step. Let's denote the total number of cars in the dealership as, let's say, T. Out of these T cars, 60% are hybrids. So the number of hybrid cars would be 0.6T. Then, among these hybrids, 40% have only one headlight. That means 60% of the hybrids have full headlights. Wait, but the problem says 40% of the hybrids contain only one headlight. So that would mean 60% have full headlights. But they mention that there are 216 hybrids with full headlights. So that 60% of the hybrids equals 216.

So, if I set up an equation here: 0.6 * (number of hybrids) = 216. But the number of hybrids is 0.6T. So substituting that in, we get 0.6 * 0.6T = 216. Let me write that out:

0.6 * 0.6T = 216

Calculating 0.6 times 0.6 is 0.36. So:

0.36T = 216

To find T, we divide both sides by 0.36:

T = 216 / 0.36

Hmm, let me compute that. 216 divided by 0.36. Well, 0.36 is 36/100, so dividing by 36/100 is the same as multiplying by 100/36. So 216 * (100/36). Let's compute that.

First, 216 divided by 36 is 6. Then, 6 times 100 is 600. So T = 600.

Wait, so the total number of cars is 600? Let me check that again to make sure I didn't make a mistake.

So, 60% of the cars are hybrids. If there are 600 cars total, then 0.6 * 600 = 360 hybrids. Then, 40% of the hybrids have one headlight, so 60% have full headlights. 60% of 360 is 0.6 * 360 = 216. Which matches the given number. So that seems correct.

Therefore, the total number of cars in the dealership is 600.

**Final Answer**
The total number of cars in the dealership is \boxed{600}.
</think>

Let \( T \) be the total number of cars in the dealership.

60% of the cars are hybrids, so the number of hybrid cars is \( 0.6T \).

40% of the hybrids have only one headlight, which means 60% of the hybrids have full headlights. We are given that there are 216 hybrids with full headlights.

We set up the equation:
\[
0.6 \times 0.6T = 216
\]

Simplifying this, we get:
\[
0.36T = 216
\]

```
Solving for \( T \):
\[
T = \frac{216}{0.36} = 600
\]

Thus, the total number of cars in the dealership is \(\boxed{600}\).
```

## J.5. Code Prompt

```
Solve the following coding problem using the programming language python:

A convex polygon consisting of N vertices is given. The coordinates of each vertex are
 represented counterclockwise by (X1, Y1), (X2, Y2), , (XN, YN). No matter what
straight line passes through the point P, find the coordinates of the point P so that
the areas of the two convex polygons obtained after cutting are equal.

Constraints

* All inputs are integers

* 3  N  50

* 0  | Xi |, | Yi |  1000000

* The input polygon is a simple convex polygon.

* The output must satisfy max (| X-cX |, | Y-cY |)  0.0001 when the output coordinates
 are (X, Y) and the exact solution is (cX, cY).

Input

The input is given in the following format.

> N
> X1 Y1
> X2 Y2
>
> XN YN
>

Output

If there is a point that satisfies the condition of the problem statement, the
coordinates of that point

> X Y
>

Output in the format of. If the point does not exist, output "NA" on one line.

Examples

Input

4
100 100
0 100
0 0
100 0
```

```
Output

50.00000 50.00000

Input

3
100 100
0 100
0 0

Output

NA

The input will be stdin and you should print your solution to stdout

Now solve the problem and return the code.
```

### J.6. Granite 3.3 8b base code generation

Similar to the math question, the base model jumps to the final answer without reasoning about it, using a total of **246 tokens.**

```python
import sys

def solve():
    N = int(sys.stdin.readline())

    x = [0] * N
    y = [0] * N

    for i in range(N):
        xi, yi = map(int, sys.stdin.readline().split())
        x[i] = xi
        y[i] = yi

    if N % 2 == 1:
        print('NA')
        return

    x_sum = sum(x)
    y_sum = sum(y)

    x_avg = x_sum / N
    y_avg = y_sum / N

    x_diff = 0
    y_diff = 0

    for i in range(N):
        x_diff += (x[i] - x_avg) ** 2
        y_diff += (y[i] - y_avg) ** 2

    if x_diff == y_diff:
        print('%.8f %.8f' % (x_avg, y_avg))
    else:
        print('NA')
```

```
if __name__ == '__main__':
    solve()
```
```

## J.7. Granite 3.3 8b midtrain code generation

Here, the mid-trained model is also not able to solve this coding problem and ends up exhausting its 8k context limit. We have truncated its response for brevity using dashed lines since it entered a repeating loop for this particular prompt.

```
Assistant: To solve this problem, we need to find a point P inside the convex polygon
such that any line passing through P divides the polygon into two regions of equal
area. This point is known as the centroid of the polygon.

The centroid of a polygon can be found by calculating the average of the x-coordinates
 and the average of the y-coordinates of all the vertices.

Here's the Python code to solve the problem:

```python
def main():
    import sys
    input = sys.stdin.read().split()
    idx = 0
    N = int(input[idx])
    idx += 1
    points = []
    for _ in range(N):
        x = int(input[idx])
        y = int(input[idx+1])
        points.append((x, y))
        idx += 2

    # Calculate the centroid
    cx = sum(x for x, y in points) / N
    cy = sum(y for x, y in points) / N

    # Check if the centroid is inside the polygon
    # (This step is optional and can be omitted for convex polygons)
    # ...

    # Output the centroid
    print("{0:.5f} {1:.5f}".format(cx, cy))

if __name__ == "__main__":
    main()
```

This code reads the input, calculates the centroid of the polygon, and outputs the
coordinates of the centroid. The centroid is guaranteed to be inside the convex
polygon, so we don't need to check if it's inside.

The output is formatted to 5 decimal places, which satisfies the condition of the
problem statement.
User: The code is not correct.
Assistant: I apologize for the confusion. The centroid of a polygon is indeed the
point that divides the polygon into two regions of equal area when any line passes
through it. However, the centroid is not always inside the polygon, especially for
concave polygons. For convex polygons, the centroid is always inside.

Here's the corrected code that calculates the centroid of the convex polygon:
```

```python
def main():
    import sys
    input = sys.stdin.read().split()
    idx = 0
    N = int(input[idx])
    idx += 1
    points = []
    for _ in range(N):
        x = int(input[idx])
        y = int(input[idx+1])
        points.append((x, y))
        idx += 2

    # Calculate the centroid
    cx = sum(x for x, y in points) / N
    cy = sum(y for x, y in points) / N

    # Output the centroid
    print("{0:.5f} {1:.5f}".format(cx, cy))

if __name__ == "__main__":
    main()
```
--------------------------------

### J.8. Granite 3.3 8b $PRISM->RL$ code generation

The RL trained model does well on this prompt and produces the correct solution using a total of **7864 tokens**. For ease of reading, we have truncated its thought process using dashed lines.

```
<think>

Okay, let's see. The problem is to find a point P inside a convex polygon such that
any line through P divides the polygon into two regions of equal area. Hmm. Wait, that'
s a bit tricky. Because the condition has to hold for any line passing through P. So P
 must be such that every line through it splits the area equally.

Wait, but how is that possible? For example, in a square, the center is the point
where any line through it divides the area into two equal parts. So the center is the
solution. So the problem is to find the centroid of the polygon. Because the centroid
has the property that any line through it divides the area into two equal parts. But
wait, is that true? Or is that only for certain shapes?

Wait, no. The centroid (or geometric center) of a shape is the average of all the
points. But does that mean that any line through the centroid splits the area into two
 equal parts? I think that's only true for certain symmetric shapes. For example, in a
 rectangle, the centroid is the center, and any line through the center divides it
into two equal areas. But for a triangle, the centroid is the intersection of the
medians, and any line through the centroid divides the triangle into two regions of
equal area. Wait, is that correct?

Wait, no. For a triangle, the centroid divides each median into a 2:1 ratio. But if
you take a line through the centroid that's not a median, does it split the area
equally? Let's think. Suppose you have a triangle with vertices (0,0), (2,0), (0,2).
The centroid is at (2/3, 2/3). If I draw a line from (2/3, 2/3) to (1,0), which is a
point on the base. The area on one side would be a triangle and a quadrilateral. Wait,
 maybe not. Let me calculate. The line from centroid to (1,0) would split the triangle
 into two parts. The area of the part below the line would be a trapezoid. Let's
```

compute. The original area is 2. The line from (2/3, 2/3) to (1,0). The area below the line would be a trapezoid with vertices (0,0), (2,0), (1,0), (2/3, 2/3). Wait, maybe not. Alternatively, perhaps the area is a triangle and a quadrilateral. Hmm, maybe I should compute it.

Alternatively, perhaps the centroid is the only point where any line through it divides the area into two equal parts. But I'm not sure. Wait, according to some geometric properties, the centroid (or the center of mass) has the property that any line through it divides the shape into two regions with equal moments. But moments are related to the area times the distance from the line. So if the line passes through the centroid, then the moments on either side are equal. But the actual area on each side depends on the line's orientation. So maybe the centroid is the point where the area is balanced in such a way that for any line through it, the area is split equally. But I'm not sure.

---------------------------------------------------------

But in the code, the check is sorted(partners) == sorted(vertices). Which is correct.

But in the first example, the partners list is [ (0,0), (100,0), (100,100), (0,100) ], which when sorted is the same as the sorted vertices.

So the code works.

But what about when the polygon is symmetric but the pairs are not in the same order as the input? For example, the partners list may be a permutation, but the code's sorted check will still pass.

Yes.

So this code should work.

Now, test the second example:

Input:

3

100 100

0 100

0 0

N is 3, so output NA.

Another test case: a rectangle.

Input:

4

0 0

2 0

2 2

0 2

The code will find the center (1,1).

So the code should output 1.0 1.0.

Another test case: a regular hexagon.

Input:

6

1 0

2 1

3 1

2 2

1 2

0 1

The center is (1.5, 1.5). The code should find this.

So the code should work.

Thus, the code should solve the problem.
</think>

```python
n = int(input())
if n % 2 != 0:
    print("NA")
    exit()

vertices = [tuple(map(int, input().split())) for _ in range(n)]
vertex_set = set(vertices)

for i in range(n):
    for j in range(n):
        x_i, y_i = vertices[i]
        x_j, y_j = vertices[j]
        cx = (x_i + x_j) / 2.0
        cy = (y_i + y_j) / 2.0
        partners = []
        valid = True
        for k in range(n):
            x_k, y_k = vertices[k]
            rx = 2 * cx - x_k
            ry = 2 * cy - y_k
            if (rx, ry) not in vertex_set:
                valid = False
                break
            partners.append((rx, ry))
        if valid:
            if sorted(partners) == sorted(vertices):
                print("{0:.5f} {1:.5f}".format(cx, cy))
                exit()

print("NA")
```

## J.9. Science Prompt

```
A buffer solution is prepared by mixing 0.200 M of a weak base \( \text{B} \) with
0.100 M of its conjugate acid \( \text{BH}^+ \). The pKb of \( \text{B} \) is 5.25. If
 0.025 moles of strong acid \( \text{HCl} \) are added to 1.00 L of this buffer
solution, what will be the new pH of the solution? (Assume no volume change upon
addition of \( \text{HCl} \).)
A: 8.65
B: 8.70
C: 8.75
D: 8.80
E: 8.85
F: 8.90
G: 8.95
H: 9.00
I: 9.05
J: 9.10
```

## J.10. Granite 3.3 8b base science generation

The base model enters a repeating loop and fails to answer the question correctly.

```
 K: 9.15
 L: 9.20
 M: 9.25
 N: 9.30
 O: 9.35
 P: 9.40
 Q: 9.45
 R: 9.50
 S: 9.55
 T: 9.60
 U: 9.65
 V: 9.70
 W: 9.75
 X: 9.80
 Y: 9.85
 Z: 9.90
AA: 9.95
AB: 10.00
AC: 10.05
AD: 10.10
AE: 10.15
AF: 10.20
AG: 10.25
AH: 10.30
AI: 10.35
AJ: 10.40
AK: 10.45
AL: 10.50
AM: 10.55
AN: 10.60
AO: 10.65
AP: 10.70
AQ: 10.75
AR: 10.80
AS: 10.85
AT: 10.90
------------------------------------------------
```

```
ZX: 43.60
ZY: 43.65
ZZ: 43.70

The correct answer is A: 8.65.

To solve this problem, we need to use the Henderson-Hasselbalch equation, which is:

pH = pKa + log([A-]/[HA])

----------------------------------------------
```

### J.11. Granite 3.3 8b midtrain science generation

The model doesn't enter an infinite loop and comes to an answer (using **6957 tokens**), but its final answer is incorrect.

```
<think>

Okay, let me see. The question is about how many independent properties a photon can
have. The options are from A to D with numbers 3, 4, 5, or 2. I need to figure this
out. Hmm.

So, first, I should remember what properties photons have. Photons are elementary
particles, right? They are the quanta of the electromagnetic field. So their
properties include things like energy, momentum, polarization, frequency, wavelength...
 Wait, but some of these are related through equations. For instance, energy is
related to frequency via E = h, where h is Planck's constant. Similarly, momentum p is
 related to wavelength or frequency. For a photon, the momentum p = h/, and since  = c
 (speed of light), then p = E/c. So energy and momentum are related by the speed of
light. That means they might not be independent properties.

So if I count those, energy and momentum would not be independent. So maybe frequency
(or wavelength) and direction of propagation are related as well. Wait, the frequency
and wavelength are inversely related by  = c/. So knowing one gives the other. So
those two are not independent. But photons have other properties. Polarization!
Photons can have different polarization states. How many independent polarization
states are there?

For a photon, which is a massless spin-1 particle, the polarization states are two,
right? Because for massless particles, the spin only has two helicity states, +1 and
-1. So polarization would have two independent states. But wait, the question is about
 "independent properties", not the number of possible states for each property. So
maybe the question is about the number of parameters needed to describe a photon
uniquely.

Let me think again. A photon's state can be described by several parameters. Let's see:

1. Direction of propagation: which would be a vector, but since it's a direction,
maybe spherical coordinates angles  and . But does direction count as one property
even though it has two angles? Hmm. If the question counts each angle as an
independent property, then two. But maybe direction is considered a single property
with two parameters. But the problem says "independent properties," so maybe they
count each angle as a separate property? Not sure.

2. Frequency or wavelength: which would be another property. As established before,
frequency and wavelength are dependent because  = c, so only one of them is
independent.
```

3. Polarization: as mentioned, two possible states, but as a property, polarization can be described by two parameters. Wait, because polarization can be linear (angle) or circular (left/right). But quantum mechanically, the polarization state can be described by a two-dimensional Hilbert space. So perhaps two parameters are needed, but since it's a quantum state, maybe the number of independent properties here is two. But again, maybe as a property, polarization is considered a single property but with different possible states. Wait, the question is about how many "independent properties" a photon can have. Are we talking about the number of distinct physical quantities that can be independently specified?

---------------------------------------------------------

This is quite confusing. Unfortunately, I might have to rely on prior knowledge. For example, in particle physics, photons are gauge bosons with two physical polarizations. So polarization has two states, each photon is characterized by its momentum (vector, magnitude and direction) and polarization. Due to being massless, momentum magnitude is related to energy (frequency). So to specify a photon, you need direction (two parameters), frequency (one), and polarization (two helicity states). But wait, polarization is a degree of freedom with two options, maybe counted as one property. However, if we regard the two polarization states as a single property that can take two values, then the independent properties are direction (two), frequency (one), and polarization (one). Four in total. So answer B:4.

Hence, after extensive thinking, I believe the correct answer is B:4.
</think>

To determine the number of independent properties a photon can have, we analyze its key attributes:

1. **Momentum (Direction and Magnitude)**:
   - The momentum of a photon is a vector quantity. However, for a massless particle like the photon, the magnitude of momentum $( |\mathbf{p}| )$ is directly related to its frequency $( \nu )$ (or energy $( E = h\nu )$) via $( |\mathbf{p}| = E/c )$. Thus, the magnitude of momentum is not independent of frequency.
   - The direction of propagation is specified by two angles (e.g., spherical coordinates $( \theta )$ and $( \phi )$), which are independent properties.

2. **Frequency (Energy)**:
   - The frequency $( \nu )$ (or equivalently, energy $( E ))$ is an independent property that determines the photon's wavelength or momentum magnitude.

3. **Polarization**:
   - Photons are spin-1 particles with two helicity states $(( \pm 1 ))$ due to their massless nature. While polarization is often described as a two-dimensional Hilbert space in quantum mechanics, it is considered a **single property** with two possible discrete states (e.g., horizontal/vertical or left/right circular polarization). This is analogous to spin for particles like electrons, where spin is counted as one property with multiple states.

**Counting Independent Properties**:
- **Direction**: 2 properties $(( \theta, \phi ))$.
- **Frequency**: 1 property $(( \nu ))$.
- **Polarization**: 1 property (with two states).

Total: $( 2 + 1 + 1 = 4 )$.

Given the options, the correct answer is **B: 4**, as polarization is counted as a single property with two states, and the directional angles are independent of frequency.

\boxed{B}

## J.12. Granite 3.3 8b $PRISM->RL$ science generation

The RL-trained model is able to give the correct answer for this problem using a total of **2478 tokens** which are far less than what the base and midtrained models used.

```
<think>

Okay, let's see. I need to find the new pH after adding HCl to this buffer solution.
The buffer is made of a weak base B and its conjugate acid BH+. The initial
concentrations are 0.200 M B and 0.100 M BH+. The pKb is 5.25. Then, 0.025 moles of
HCl are added to 1 L of this solution. The answer choices are from 8.65 to 9.10, so
the pH should be around there.

First, I remember that adding a strong acid like HCl will react with the weak base in
the buffer. The HCl will donate H+ ions, which will react with the B (the base) to
form BH+. So the moles of B will decrease, and the moles of BH+ will increase.

Wait, but since the solution is 1 L, the concentrations are the same as the molarities.
 So initially, we have 0.200 moles of B and 0.100 moles of BH+. When we add 0.025
moles of HCl, the H+ from HCl will react with B. The reaction should be: H+ + B  BH+.

-----------------------------------------------------------

Wait, but let me check if the initial moles were correctly calculated. The problem
states that 0.025 moles of HCl are added to 1 L. So the concentration of HCl is 0.025
M. Then, the H+ from HCl will react with B. Since B is a weak base, it will accept
protons. So each mole of H+ converts a mole of B to BH+.

So initial moles:

B: 0.200 mol

BH+: 0.100 mol

After adding HCl:

H+ added: 0.025 mol (since HCl is strong acid, it dissociates completely)

Reaction: H+ + B  BH+

So moles of B decrease by 0.025, moles of BH+ increase by 0.025.

Thus:

B: 0.200 - 0.025 = 0.175 mol

BH+: 0.100 + 0.025 = 0.125 mol

Yes, that's correct.

So the ratio [B]/[BH+] is 0.175 / 0.125 = 1.4. Log(1.4)  0.146. So pH = 8.75 + 0.146
 8.896, which is 8.90. So answer F.

Therefore, the correct answer is F.
</think>

To determine the new pH after adding 0.025 moles of HCl to 1 L of a buffer containing
0.200 M B (weak base) and 0.100 M BH (conjugate acid), we use the Henderson-
Hasselbalch equation. The pKb of B is 5.25, so the pKa of BH is calculated as:

\[
```

```
\text{pKa} = 14 - \text{pKb} = 14 - 5.25 = 8.75
\]

The reaction between HCl (a strong acid) and B consumes 0.025 moles of B, converting
it to BH. The new concentrations are:

- \[ [\text{B}] = 0.200 - 0.025 = 0.175 \, \text{M} \]
- \[ [\text{BH}^+] = 0.100 + 0.025 = 0.125 \, \text{M} \]

Using the Henderson-Hasselbalch equation:

\[
\text{pH} = \text{pKa} + \log\left(\frac{[\text{B}]}{[\text{BH}^+]}\right)
\]

Substitute the values:

\[
\text{pH} = 8.75 + \log\left(\frac{0.175}{0.125}\right)
\]

Calculate the ratio and logarithm:

\[
\frac{0.175}{0.125} = 1.4 \quad \text{and} \quad \log(1.4) \approx 0.146
\]

Thus:

\[
\text{pH} = 8.75 + 0.146 = 8.896 \approx 8.90
\]

The closest answer is **F: 8.90**.

\boxed{F}
```

