# OpenReview forum: "PRISM: Demystifying Retention and Interaction in Mid-Training"
_ICML.cc/2026/Conference — ICML 2026 spotlight_

### Official Review · Reviewer_nzt4 · 2026-03-02

**Soundness:** 3
**Presentation:** 2
**Significance:** 3
**Originality:** 4
**Overall Recommendation:** 5
**Confidence:** 4

**Summary:**

The paper introduces a new framework PRISM that provides a data recipe for LLM mid-training by considering retaining general knowledge and long-context capabilities learned in pre-training, broad domain coverage and how mixing data from different domains affect domain-specific benchmarking results, and the enablement of RL on verifiable domains. The authors validated their recipe on various model families e.g., Granite, Mistral, Llama; sizes e.g., 3B, 7B, 8B, 24B; and architectures e.g., dense, hybrid layers, MoE.

**Compliance With Llm Reviewing Policy:**

Affirmed.

**Final Justification:**

In the rebuttal, I found authors' explanation on my W1 interesting, especially on later RL gains. The mechanistic analysis is useful, and the removal of web data ablation fully resolves my W3.

I remain my opinion that the paper lacks technical novelty (even with the mechanistic analyses, as these are still empirical insights), but given the already existing deep empirical insights this manuscript presents, I am happy to raise my score to 5.

**Key Questions For Authors:**

1. I don't understand why as the midtraining on math+code goes, code avg achieves the strongest performance at 10.49B tokens but consistently declines as mode math+code tokens are used. Could authors provide any explanations for that?
2. In L256 authors mentioned that the attribution of performance maintenance on general benchmarks are partially the inclusion of general web data, but I wonder what all the numbers would be if such data is not included i.e., baseline, as well as the scaling/proportional property of including such data in the mid-training data mixture?

**Limitations:**

The authors had a short discussion on impact statement but should also include environmental impact such as the amount of electricity e.g., GPU hours used for all experiments. There is no mentioning of the limitation. As a pure empirical study, albeit having validated on different models, the conclusions might not generalize to every model; and as time goes (e.g., different pretraining strategy emerges; post-training needs change), the recipe can get outdated and thus provide less help.

**Strengths And Weaknesses:**

### Strengths
* The recipe for PRISM is quite comprehensive and covered multiple aspects that are often ignored or incompletely reported in prior works e.g., pretrained capability retention, domain and evaluation coverage and RL compatibility.
* Many experiments are well-designed, such as controlling the token budget in the domain data mixture experiments, where authors showed synergetic improvements across domains. Further, these are validated across different model families, sizes, and architectures.
* The ablation studies are useful, in particular how to effectively restore long-context ability after non-long-context mid-training and how performance in different benchmarks changes as models are mid-trained. While there is little technical novelty, the empirical contribution especially regarding the cost of such training is helpful to the community.

### Weaknesses
* While PRISM appears to be a successful mid-training recipe for their main experimentation model Granite-3.3 (8B), whether such success will always apply to other models remains unclear. For example, both Mistral models had a performance drop in the Science domain after the Math+Code+Science mid-training; the fluctuation on general benchmarks is big across different models where some models benefit from such mid-training but other models don't.
* The presentation can sometimes lead to confusion. For example, L130 has a mentioning of "Qwen2.5-100M" without reference and it is not a commonly known model (e.g., it was a pretrained model from a specific paper). There are missing references in the first place e.g., L44 "LB-V1/V2, Ruler, LiveCodeBench, Codeforces, AIME, MATH500, and GPQA-Diamond." Figure 2 can be better presented with another hue that captures domains. Many figures could have bigger fonts.
* There is not really anything technically new in this paper, where retaining pretraining knowledge by incorporating pretraining, long-context annealing, merging, and RL baselines are all existing techniques, though the empirical findings are still useful.
* The paper could have added more experiments to explore some interesting patterns; for example the authors did not provide any further experimentation into why code performance initially jumped high but consistently declines as mode math+code tokens are used; the authors also did not experiment with the amount of general web data (and perhaps how such data) in order to retain pretrained capabilities, which is one of the most important selling point of PRISM.

---

> ### Author Rebuttal · Authors · 2026-03-30
>
> We thank Reviewer nzt4 for the thorough review, recognition of our empirical contribution. We address each concern below, including new mechanistic analyses :
>
> **W1: Inconsistency across models (Mistral science drop, general fluctuation).**
>
> The Mistral science drop at mid-training is expected, reflecting differences in pretraining distributions. Mid-training impact depends on alignment between pretraining and mid-training data.
>
> Importantly, science capability emerges strongly during RL: Mistral-Small-24B achieves +27.95 GPQA-Diamond after RL (Figure 4), the largest science gain in the study. This shows science data at mid-training seeds latent capability that RL unlocks, even when mid-training gains appear modest.
>
> General benchmark fluctuation reflects real pretraining differences. No model shows systematic LB-V1 regression (Table 5), though individual benchmarks shift. We will clarify that PRISM provides consistent gains, but gain profiles depend on pretraining distribution.
>
> Mechanistically, CKA analysis (Granite-3.3, LLaMA-3.1) shows base→mid-training divergence is model-dependent (e.g., Granite final layer GSM8K ~0.63 vs LLaMA C4 ~0.71), while MT→RL CKA >0.998 across all models/inputs. Thus mid-training reshapes representations differently per model, while RL consistently preserves mid-training geometry.
>
> **W2: Presentation issues.**
>
> We will address the following:
> - L130 Qwen2.5-100M refers to [1] (cited at L128)
> - Missing benchmark citations will be added (LB-V1/V2, RULER, LiveCodeBench, Codeforces, AIME, MATH500, GPQA-Diamond)
> - Figure 2 will include domain-level hue encoding
> - Font sizes will be increased
>
> **W3: No technical novelty.**
>
> We respectfully note that the submission's contribution is explicitly empirical. That said, we have strengthened the technical contribution by conducting new mechanistic analyses on the already-trained models from the submission. These require no additional training, only post-hoc analysis of the existing checkpoints:
>
> **1. Weight divergence (Granite-3.3 8B):**
>
> | Transition     | Attn L2 | MLP L2 | Total L2 | Sparsity |
> |---------------|--------:|-------:|---------:|---------:|
> | Base → MT     | 0.175   | 0.329  | 0.175    | 9.3%     |
> | MT → RL       | 0.0003  | 0.0006 | 0.0003   | 95.9%    |
> | Base → RL     | 0.0004  | 0.0007 | 0.0004   | 96.0%    |
>
> Mid-training restructures >90% of parameters; RL modifies ~5% with ~580× smaller L2. RL’s footprint is starting-point invariant (same L2/sparsity/sub-components), consistent with earlier results.
>
> **2. CKA similarity (Granite-3.3, LLaMA-3.1):**
>
> | Model         | Input     | Min MT vs RL CKA |
> |--------------|-----------|------------------|
> | Granite-3.3  | Wikipedia | 0.9999           |
> | Granite-3.3  | GSM8K     | 0.9997           |
> | LLaMA-3.1    | Wikipedia | 0.9993           |
> | LLaMA-3.1    | GSM8K     | 0.9994           |
>
> MT→RL similarity >0.998 across models/inputs: RL operates within mid-training geometry. In contrast, Base→MT CKA drops as low as 0.63, confirming mid-training restructures representations.
>
> **3. Weight direction (Granite-3.3):**
>
> | Metric                          | MC   | MCS  |
> |---------------------------------|------|------|
> | Total L2 divergence from base   | 0.177| 0.175|
> | Cosine similarity (MC vs MCS)   | 0.521|      |
>
> Nearly identical magnitudes but cosine similarity of only 0.52, confirming that data composition determines the direction of model change in weight space, not just its magnitude. This mechanistically explains why MC and MCS produce different capability profiles despite similar-scale updates.
>
> Together, these provide a joint weight and representation-level characterization of mid-training vs RL at scale.
>
> **W4 / Q1: Code performance peaks then declines.**
>
> This reflects domain competition under fixed sampling ratios (Figure 2). At higher token budgets, math data dominates, diluting code-specific signal.
>
> Crucially, code at 27B remains far above base (8.56 vs 1.26, Table 5), so this is a relative decline, not regression. This motivates balanced Math+Code+Science mixtures.
>
> **Q2: Effect of removing web data.**
>
> As shown in our response to Reviewer iRjq W3, removing DCLM-EDU drops LB-V1 Avg 66.48→64.64 (below base 66.15), with largest drop on TruthfulQA (-3.64), while LB-V2 remains unchanged (~23.0). Thus web data supports general capability retention, while domain data drives reasoning gains.
>
> **Limitations and environmental impact.**
>
> We will add a Limitations section:
> - Generalizability: validated on 3B–24B across 4 families
> - Temporal validity: The specific mixture ratios may need recalibration, though we expect the general principles (balanced mixtures, retention-aware design) to remain applicable
>
> We will also report GPU usage: mid-training (25k steps, 8k context, H100), RL (1k steps), with aggregate GPU-hour estimates.
>
> [1] Zhang et al. “On the Interplay of Pre-Training, Mid-Training, and RL on Reasoning Language Models.”  (2025)

---

> > ### Author Rebuttal · Reviewer_nzt4 · 2026-04-03
> >
> > Thanks for the detailed response! I found authors' explanation on my W1 interesting, especially on later RL gains. The mechanistic analysis is useful, and the removal of web data ablation fully resolves my W3.
> >
> > I remain my opinion that the paper lacks technical novelty (even with the mechanistic analyses, as these are still empirical insights), but given the already existing deep empirical insights this manuscript presents, I am happy to raise my score to 5.

---

### Official Review · Reviewer_5cgC · 2026-03-03

**Soundness:** 3
**Presentation:** 3
**Significance:** 3
**Originality:** 3
**Overall Recommendation:** 4
**Confidence:** 2

**Summary:**

This paper introduces PRISM, a framework for analyzing "mid-training". The work demonstrates that a relatively small, high-quality mid-training phase (~27B tokens) acts as a critical "stabilizer" for reasoning capabilities. This stage significantly boosts performance in math and coding while retaining general abilities, and most importantly, it prepares the model to be much more receptive to downstream Reinforcement Learning (RL).

**Compliance With Llm Reviewing Policy:**

Affirmed.

**Key Questions For Authors:**

1. There appears to be a lack of a detailed data filtering process, with only partial information provided in Appendix A.
2. Are the ratios of different sources in a mixed dataset sensitive to performance improvements?
3. What's the improvement on other types of benchmark, such as humanities, social sciences, and even creative writing.

**Limitations:**

yes

**Strengths And Weaknesses:**

# Strengths:
1. The work empirically construct a mid-training dataset, called PRISM, including ~27B-token high-quality data covering different perspectives.
2. PRISM is validated through mid-training on different benchmarks, covering general LLM ability, long-context behavior, domain-specific reasoning, etc. The experiment results demonstrate varying degrees of improvement across different settings.
3. Verify the necessity of mid-training stage and synergy between math and code through experiments.


# Weaknesses:
1. Mid-training performance is highly sensitive to data composition. However, there are no theory analysis or summarized guideline for effectively mixing data from different sources or domains.
2. Limited generalizability regarding data scale and mixtures in mid-training. The exploration was conducted on models ranging from 3B to 23B parameters. Since this was only empirically verified, it's challenging to confirm its applicability to other scales and model architectures.
3. In addition to data mixture, data curation is crucial in dataset construction. Since data curation for most selected datasets is conducted by other works, scaling up PRISM for training larger LLMs becomes challenging.

---

> ### Author Rebuttal · Authors · 2026-03-30
>
> We thank Reviewer 5cgC for recognizing the breadth of our evaluation and the practical value of the PRISM framework. We address each concern below.
>
> **W1: No theory or summarized guideline for data mixing.**
>
> We agree that a consolidated guideline would strengthen the paper. We will add a practical guidelines box summarizing the empirically validated principles:
>
> - Math+Code+Science outperforms domain-specific mixtures (Table 4: Math Avg 48.75 and GPQA 29.12 for MCS vs 36.43 and 17.34 for Math-only)
> - Science data at mid-training seeds downstream RL gains, adding +17 to +28 GPQA-Diamond points during RL even when the RL mix is unchanged (Table 16)
> - General web data preserves general capabilities, with no LB-V1 regression across any of the 7 models (Table 5). A new leave-one-out ablation confirms this: removing DCLM-EDU web data from the MCS mixture (redistributing weights to domain sources) drops LB-V1 Avg from 66.48 to 64.64 on Granite-3.3, below the base model (66.15), while LB-V2 domain reasoning is preserved (see Reviewer iRjq W3 for full table)
> - The RL data mix matters less than the mid-training mix, with RL mix changes producing <2 point AVG differences (Table 16, Step 4)
>
> Additionally, our mechanistic analysis explains why data composition matters: cosine similarity between MC (Math+Code) and MCS (Math+Code+Science) mid-training updates on Granite-3.3 is only 0.52, despite nearly identical L2 magnitudes (0.177 vs 0.175). Thus, they steer the model in different directions in weight space despite similar scales, explaining the large downstream differences.
>
> A formal theory of data mixing would require understanding interactions between pretraining distributions and mid-training gradients across architectures, which is beyond the scope of this work.
>
> **W2: Limited generalizability (3B to 24B).**
>
> We acknowledge this limitation and will state it explicitly. However, consistency across our tested range is informative: gains hold from Granite-4 Micro (3B) to Mistral-Small (24B), across dense Transformers and hybrid architectures, and across 4 model families with different pretraining distributions. The token scaling ablation (Section 7.2) shows gains saturate by ~26B tokens even at 3B scale, suggesting the 27B budget is not scale-dependent. We will frame validation at 70B+ as future work.
>
> **W3: Data curation relies on other works; scaling up is challenging.**
>
> This is by design. A key contribution of PRISM is showing that publicly available, already-curated datasets can be combined into effective mid-training mixtures without custom curation pipelines. All datasets used (DCLM-EDU, OpenMathInstruct, RefinCode, StarCoder2, OpenThoughts3, etc.) are publicly released with their own quality controls. Our contribution is the mixture recipe and systematic analysis of how composition choices propagate through the pipeline. Reliance on open datasets improves reproducibility. For scaling, the same sources can be used at higher token budgets; our scaling analysis suggests gains saturate at moderate budgets regardless of model size.
>
> **Q1: Lack of detailed data filtering process.**
>
> We will expand Appendix A. To clarify: general web data uses DCLM-EDU filtered at quality score >= 3 per dataset guidelines. Domain reasoning datasets (OpenMathInstruct, RefinCode, StarCoder2, OpenThoughts3) use upstream curation. For RL prompt filtering, we sample 16 responses per prompt from the mid-trained model and retain prompts with exactly 1/16 correct for math (19k hard prompts) and at least 1/16 correct for code and science (7k and  17k). We will consolidate this description.
>
> **Q2: Are source ratios sensitive to performance?**
>
> Yes. Section 5 and Figure 2 show three configurations (Math-only, Math+Code, Math+Code+Science) with empirically tuned per-source sampling weights. Ratios matter: adding science without rebalancing would dilute math/code. The configurations in Figure 2 provide the best balance and are used across experiments. A full sensitivity sweep would require many runs; we will note this as future work, potentially with smaller proxy models.
>
> **Q3: Improvement on humanities, social sciences, creative writing.**
>
> Our evaluation includes LB-V1 (MMLU: humanities/social sciences/STEM; ARC: science; HellaSwag: commonsense; TruthfulQA: factuality) and LB-V2 (BBH, MUSR, MMLU-Pro). Across all 7 models, LB-V1 averages are preserved within ~1 point of base (Table 5), indicating no degradation on broad capabilities. We do not evaluate creative writing and will acknowledge this gap. Our focus is reasoning enhancement with retention, and the benchmark suite is designed to detect regressions on general knowledge and reasoning.

---

> > ### Author Rebuttal · Reviewer_5cgC · 2026-04-01
> >
> > Thank you to the authors for their response. They addressed most of my questions. I hope future work includes summarized guidelines or theoretical analysis.

---

> > > ### Author Response · Authors · 2026-04-04
> > >
> > > Thank you again for your thoughtful feedback and for acknowledging that your concerns have been addressed. If you feel the rebuttal has resolved your questions, we would be grateful if you could consider updating your score and/or confidence to reflect this. We understand scoring is subjective, and would appreciate any consideration of whether your updated assessment might be reflected in the final score or confidence.
> > >
> > > Thank you again for your consideration.

---

### Official Review · Reviewer_iRjq · 2026-03-10

**Soundness:** 2
**Presentation:** 3
**Significance:** 3
**Originality:** 2
**Overall Recommendation:** 5
**Confidence:** 3

**Summary:**

This paper presents PRISM, a broad empirical study of mid-training for reasoning LLMs, with a focus on retention of general abilities, long-context behavior, domain-mixture effects, and compatibility with downstream RL. The authors evaluate across several model families, including Granite, LLaMA, and Mistral, and report that a relatively modest mid-training phase of about 27B high-quality tokens can substantially improve coding and math reasoning while largely preserving general performance. They further show that applying RL on top of PRISM-mid-trained checkpoints is much more effective and stable than applying RL directly to base models.

**Compliance With Llm Reviewing Policy:**

Affirmed.

**Final Justification:**

Overall, I appreciate the authors’ efforts and the clarity of the additional evidence. Based on this, I am happy to increase my score from weak accept to accept. I encourage the authors to further strengthen the final version by clarifying the data curation pipeline and explicitly detailing the identical RL configurations.

**Key Questions For Authors:**

1. How much of the reported benefit comes from general web retention data versus instruction data versus domain reasoning data? I feel single-domain and leave-one-out mixture ablations would help clarify whether the observed gains arise from specific data sources or from their interaction. (Weakness 3)
2. The paper claims RL on PRISM is much stronger than RL on base models. Can the authors include a more direct matched comparison table with identical RL budget, reward design, prompt filtering, and sampling settings?
3. How sensitive are the findings to the exact 27B-token budget and data quality assumptions across larger models?

**Limitations:**

See above.

**Strengths And Weaknesses:**

Strengths
- The study spans multiple model families and scales, and the reported gains are substantial.
- The benchmark is strong: the paper argues that evaluating only math or code can hide regressions, and therefore includes LB-V1/V2, RULER, LiveCodeBench, Codeforces, AIME, MATH500,  and GPQA-Diamond
- The RL curves on Granite-3.3 and Mistral-24B show additional gains after PRISM, while the paper argues that direct RL on base models is much less effective. Interesting and meaningful results.


Weaknesses
- I feel some central claims are stronger than the evidence fully supports. The paper repeatedly frames PRISM as a “necessary intermediate step” for reliable reasoning enhancement and RL scaling, the current results are promising, but “necessary” feels too strong for what is ultimately an empirical study rather than a controlled proof of indispensability.
- For RL training, the paper uses GRPO and a particular reward/format setup, including a required <think> ... </think> format reward, plus prompt filtering with a mid-trained model. That means the RL gains may partly depend on several design choices beyond PRISM itself, and the paper could do more to isolate the causal role of the mid-training stage from the RL pipeline details.
- The current ablations do not fully disentangle the respective contributions of general web retention data, instruction data, and domain-specific reasoning data. Single-domain and leave-one-out mixture ablations would help clarify whether the observed gains arise from specific data sources or from their interaction.

---

> ### Author Rebuttal · Authors · 2026-03-30
>
> We thank **Reviewer iRjq** for recognizing the breadth of our evaluation and the meaningful RL results. We address each concern below.
>
> **W1: "Necessary" framing too strong.**
> We agree. We will replace "necessary intermediate step" with "empirically strong foundation" and "consistently beneficial." Our evidence is empirical: mid-training consistently improves RL outcomes across all tested models, without claiming theoretical indispensability.
>
> **W2: RL design choices not isolated from PRISM.**
> Base→RL and PRISM→RL use identical RL configurations: same GRPO, reward (<think>...</think>), sampling (16, T=1.0), prompt pool, training budget (1000 steps), and hardware. The only variable is initialization. We will clarify this in Section 8.
>
> Prompt filtering was done once using the mid-trained model, and the same filtered set was used for both Base→RL and PRISM→RL, so differences are attributable to initialization.
>
> We further ran **mechanistic experiments** (Granite-3.3, MCS checkpoint). RL produces nearly identical update patterns:
>
> | Metric                          | PRISM→RL | Base→RL |
> |---------------------------------|----------|---------|
> | Total L2 divergence             | 0.0003   | 0.0004  |
> | Attention L2                    | 0.0003   | 0.0004  |
> | MLP L2                          | 0.0006   | 0.0007  |
> | Sparsity (% <1% change)         | 95.9%    | 96.0%   |
> | V projection (% changed)        | 5.7%     | 7.5%    |
> | O projection (% changed)        | 5.6%     | 6.7%    |
> | MLP gate (% changed)            | 5.4%     | 6.1%    |
>
> Despite nearly identical weight updates, downstream performance differs drastically (Fig. 9 / Table 16):
>
> | Benchmark       | Base→RL | PRISM→RL | Gap    |
> |----------------|--------:|---------:|-------:|
> | AIME24         | 1.46    | 40.42    | +38.96 |
> | AIME25         | 0.62    | 29.58    | +28.96 |
> | MATH500        | 36.69   | 83.99    | +47.30 |
> | LiveCodeBench  | 7.53    | 17.20    | +9.67  |
> | Codeforces     | 4.86    | 18.03    | +13.17 |
> | GPQA-Diamond   | 29.80   | 51.52    | +21.72 |
>
> RL applies similar updates (magnitude, sparsity, components). The difference is the starting representation, directly isolating mid-training's contribution.
>
> **W3 / Q1: Ablations do not disentangle data sources.**
> Section 5 controls domain composition (Math, +Code, +Science at fixed 27B), showing clear contributions (e.g., +10.10 GPQA-Diamond at mid-training and +17–28 more after RL).
>
> To directly address this concern, we ran an additional leave-one-out ablation on Granite-3.3 removing general web data (DCLM-EDU) from the MCS mixture and redistributing sampling weights to the remaining domain sources:
>
> | Benchmark     | Base  | MCS (with DCLM) | MCS (no DCLM) |
> |---------------|------:|----------------:|--------------:|
> | ARC           | 61.95 | 61.69           | 59.56         |
> | HellaSwag     | 83.46 | 78.12           | 77.12         |
> | MMLU          | 62.56 | 62.98           | 62.16         |
> | TruthfulQA    | 52.24 | 46.96           | 43.32         |
> | Winogrande    | 80.35 | 74.90           | 74.51         |
> | GSM8K         | 56.33 | 74.22           | 71.19         |
> | LB-V1 Avg     | 66.15 | 66.48           | 64.64         |
> | IFEval        | 46.62 | 46.44           | 45.94         |
> | BBH           | 24.68 | 26.32           | 27.19         |
> | MATH          | 10.20 | 20.02           | 18.05         |
> | GPQA          | 6.38  | 7.27            | 8.39          |
> | MUSR          | 8.88  | 8.60            | 9.91          |
> | MMLU-Pro      | 23.82 | 29.55           | 28.53         |
> | LB-V2 Avg     | 20.10 | 23.03           | 23.00         |
>
> Removing web data drops LB-V1 Avg (66.48→64.64, below Base 66.15), with the largest drop on TruthfulQA (-3.64), while LB-V2 is largely preserved. This confirms web data mainly supports general capability retention. Instruction ablation remains future work.
>
> **Q2: Matched comparison table.**
> Included above in our W2 response; we will add it explicitly with all variables held constant.
>
> **Q3: Sensitivity to 27B tokens and scale.**
> Section 7.2 (token scaling on Granite-4 Micro 3B) shows gains emerge early (~10B tokens) and saturate by ~26B: Math Avg goes from 26.28 (base) to 40.21 (10.49B) to 42.42 (31.46B). The 27B budget sits in the saturation regime, not at a fragile operating point.
> For larger models, Mistral-Small-24B (24B, our largest) shows the same pattern: +53.88 MATH500 improvement at mid-training and the largest GPQA-Diamond RL gain of +27.95 (Figure 4). This suggests the findings extend beyond 8B models, though we will acknowledge that we have not validated at 70B+ scale.

---

> > ### Author Rebuttal · Reviewer_iRjq · 2026-04-01
> >
> > Thank you for the detailed rebuttal and for conducting additional controlled and mechanistic experiments within such a short time frame. I appreciate the care taken to isolate variables and to clarify that the observed differences are attributable to initialization rather than the RL configuration.
> >
> > Overall, I appreciate the authors’ efforts and the clarity of the additional evidence. Based on this, I am happy to increase my score from weak accept to accept. I encourage the authors to further strengthen the final version by clarifying the data curation pipeline and explicitly detailing the identical RL configurations.

---

### Decision · Program_Chairs · 2026-04-30

**Decision:**

Accept (spotlight)

**Comment:**

This is an interesting paper on understanding mid-training. The idea is to study the design choices made during mid-training and their downstream consequences. There is a very nice collection of insights here.

Reviewers liked the number of model families/sizes and the number of tasks, the resulting insights, and additionally the extra pieces brought up by the authors during the rebuttal phase.

This is a strong paper with solid methodology that provides interesting insights for the community.